# Cross-tissue transcriptome-wide association studies identify susceptibility genes shared between schizophrenia and inflammatory bowel disease

Florian Uellendahl-Werth [1], Carlo Maj [2], Oleg Borisov[2], Simonas Juzenas [1,3], Eike Matthias Wacker [1], Isabella Friis Jørgensen [4], Tim Alexander Steiert[1], Saptarshi Bej[5], Peter Krawitz [2], Per Hoffmann[6,7], Christoph Schramm[8], Olaf Wolkenhauer [5,9,10], Karina Banasik [4], Søren Brunak [4], Stefan Schreiber[1,11], Tom Hemming Karlsen [12], Franziska Degenhardt[13], Markus Nöthen [6,7], Andre Franke[1], Trine Folseraas[12] & David Ellinghaus [1,4 ✉]

Genetic correlations and an increased incidence of psychiatric disorders in inflammatory-bowel disease have been reported, but shared molecular mechanisms are unknown. We performed cross-tissue and multiple-gene conditioned transcriptome-wide association studies for 23 tissues of the gut-brain-axis using genome-wide association studies data sets (total 180,592 patients) for Crohn's disease, ulcerative colitis, primary sclerosing cholangitis, schizophrenia, bipolar disorder, major depressive disorder and attention-deficit/hyperactivity disorder. We identified *NR5A2*, *SATB2*, and *PPP3CA* (encoding a target for calcineurin inhibitors in refractory ulcerative colitis) as shared susceptibility genes with transcriptome-wide significance both for Crohn's disease, ulcerative colitis and schizophrenia, largely explaining fine-mapped association signals at nearby genome-wide association study susceptibility loci. Analysis of bulk and single-cell RNA-sequencing data showed that *PPP3CA* expression was strongest in neurons and in enteroendocrine and Paneth-like cells of the ileum, colon, and rectum, indicating a possible link to the gut-brain-axis. *PPP3CA* together with three further suggestive loci can be linked to calcineurin-related signaling pathways such as NFAT activation or Wnt.

[1] Institute of Clinical Molecular Biology, Christian-Albrechts-University of Kiel, Kiel, Germany. [2] Institute for Genomic Statistics and Bioinformatics, University of Bonn, Bonn, Germany. [3] Institute of Biotechnology, Life Science Centre, Vilnius University, Vilnius, Lithuania. [4] Novo Nordisk Foundation Center for Protein Research, Disease Systems Biology, Faculty of Health and Medical Sciences, University of Copenhagen, Copenhagen, Denmark. [5] Department of Systems Biology & Bioinformatics, University of Rostock, Rostock, Germany. [6] Department of Genomics, Life & Brain Center, University of Bonn, Bonn, Germany. [7] Institute of Human Genetics, School of Medicine, University of Bonn & University Hospital Bonn, Bonn, Germany. [8] First Department of Internal Medicine, University Medical Center Hamburg-Eppendorf, Hamburg, Germany. [9] Stellenbosch Institute of Advanced Study (STIAS), Wallenberg Research Centre at Stellenbosch University, 7602 Stellenbosch, South Africa. [10] Leibniz Institute for Food Systems Biology, Technical University Munich, Freising, Germany. [11] First Medical Department, University Hospital Schleswig-Holstein, Kiel, Germany. [12] Research Institute for Internal Medicine, Division of Surgery, Inflammatory Diseases and Transplantation, Oslo University Hospital Rikshospitalet and University of Oslo, Oslo, Norway. [13] Department of Child and Adolescent Psychiatry, Psychosomatics and Psychotherapy, University Hospital Essen, University of Duisburg-Essen, Duisburg, Germany. ✉email: d.ellinghaus@ikmb.uni-kiel.de

The direct link (gut-brain-axis [GBA]) of intestinal dysfunction and inflammation with brain development and mental illness is one of the most current topics in biomedical research[1–3]. Gastrointestinal symptoms have been observed in patients with neurobehavioral, neurodevelopmental, and mental diseases, including attention-deficit/hyperactivity disorder (ADHD)[4], schizophrenia (SCZ)[5], and major depressive disorder (MDD)[6]. Vice versa, psychiatric comorbidity, including depression, anxiety, bipolar disorder (BD), and SCZ, is known to be more prevalent in inflammatory bowel disease (IBD) patients[7–9]. Given the polygenic nature of most psychiatric and chronic inflammatory diseases, some of the (potentially shared) phenotypic variation in disease risk is expected to be explained by expression quantitative trait loci (eQTL) active in disease-relevant tissues of the GBA[10]. A significant shared proportion of genomic correlation was detected between inflammatory (Crohn's disease [CD], ulcerative colitis [UC]), and psychiatric traits (SCZ, BD) using LD score regression (LDSC)[11]. However, it remains unclear, whether the correlation is due to a few loci or is distributed across the genome.

In this study, we performed cross-tissue transcriptome-wide association studies (TWAS) for 23 tissues of the GBA using genome-wide association study (GWAS) summary statistics from ten case-control GWAS data sets (total 180,592 patients and 290,737 nonoverlapping controls, Supplementary Data 1; Methods) of three clinically related inflammatory diseases of the gastrointestinal tract (CD, UC and primary sclerosing cholangitis (PSC) because PSC patients suffer from a highly increased frequency (62–83%) of IBD called PSC-IBD[12]) and four diseases of the mind and brain (SCZ, MDD, BD, and ADHD) for which moderate to strong genetic correlation ($r_g > 0.3$) at the genome-wide level has been described between chronic inflammatory diseases[13] and psychiatric diseases[14], respectively. The TWAS approach can be viewed as a transcriptome-wide screening method to test for gene-disease associations from GWAS data sets[15]. By using a recently published principled cross-tissue TWAS method (UTMOST)[16]—simultaneously training expression imputation models across multiple disease-relevant tissues and performing cross-tissue gene-level conditioned association tests—in combination with GWAS/TWAS conditional analysis for cross-phenotype studies we were able to address several types of confounding factors that have previously compromised the validity of TWAS studies for complex diseases[15,17]: (1) poor TWAS prediction accuracy due to imputation from reference expression panels containing tissues not relevant to disease, (2) a bias in the number of transcriptome-wide significant gene-disease associations due to different sample sizes of reference data sets in expression imputation models[18], (3) an excess of TWAS association signals at loci with multiple TWAS signals due to correlated expression among genes of the same locus, and (4) an inaccurate determination of association boundaries for TWAS signals based on SNP information.

The objectives of this cross-tissue, cross-phenotype TWAS were to (i) evaluate the effectiveness of cross-tissue imputation models using ten GWAS consortium studies of psychiatric and inflammatory disease; (ii) identify transcriptome-wide significant gene-disease associations in GBA tissues for each disease and localize the most strongly associated genes through cross-tissue and multiple-gene-conditioned fine-mapping analyses; (iii) quantify the overall component of (shared) genetically regulated expressional changes ($\rho_E$) for immune-mediated and psychiatric diseases in relation to genome-wide genetic correlation ($\rho_G$); (iv) identify comorbidities and causal relationships using population-based administrative health data from the entire Danish population and by performing Mendelian Randomization (MR) analyses; (v) identify susceptibility genes and pathways shared between psychiatric disorders and inflammatory diseases of the gastrointestinal tract using bulk RNA-Seq data of different developmental stages and single-cell RNA-sequencing (scRNA-seq) expression data for tissues of the GBA.

## Results

### Cross-tissue transcriptome-wide association studies (TWAS) for seven diseases of the GBA

The UTMOST framework was used to perform cross-tissue TWAS for 17,290 genes and 23 selected disease-relevant tissues, which had been identified as disease-relevant for diseases of the GBA in a previous study of Finucane and colleagues[19] (Methods), using transcriptome-wide and genome-wide reference data of 4043 samples provided by consortia projects GTEx[20], STARNET[21], and BLUEPRINT[22] (Fig. 1 and Supplementary Data 2). A schematic overview of the study workflow is shown in Supplementary Fig. 1. Gene expression imputation models were used using GWAS summary statistics data of ten consortium meta-analysis studies comprising 180,592 independent cases of European ancestry (CD: 21,771; UC: 18,621; PSC: 4,338; SCZ: 35,476; MDD: 59,851; BD: 20,352; ADHD: 20,183; Supplementary Data 1). We then tested for gene-disease associations for a maximum of 17,290 genes and 23 tissues for each of the ten GWAS meta-analysis studies. Single-tissue gene-wise (marginal; i.e. unconditioned) association $P$ values (analysis Single-tissue$_{marginal}$; Supplementary Data 3) were combined into (marginal) joint-tissue gene-wise summary $P$ values (analysis GBJ$_{marginal}$, resulting in one $P_{GBJ\_marginal}$ for each gene and disease; Supplementary Data 3) via a generalized Berk–Jones (GBJ) test[23]. The cross-tissue GBJ gene test quantifies gene-disease associations across all tissues of the GBA (separately for each disease), adjusts the correlation between TWAS association statistics for individual tissues, and also provides (compared with standard univariate meta-analysis approaches) a substantial increase in statistical power in situations where eQTL effects are not unique to a single tissue (Methods). The GBJ test identified a total of 586 (277), 345 (179), 214 (79), 361 (104), 38 (21), 77 (10), and 42 (8) gene-disease associations (number of loci with size of ±1 Mb) with transcriptome-wide significance ($P_{GBJ\_marginal} < 0.05/15,587 = 3.20 \times 10^{-6}$; Supplementary Fig. 2; corrected for a maximum of 15,587 successfully tested and GBJ-meta-analyzed genes) for CD, UC, PSC, SCZ, MDD, BD, and ADHD, respectively (Supplementary Data 3). To evaluate a possible bias in the number of transcriptome-wide significant gene-disease associations due to different sample sizes of reference data sets, we further conducted TWAS analyses using expression imputation models provided by S-PrediXcan[24] and FUSION[25] (Methods). We observed on average only a moderate positive correlation between reference sample size and the number of significant genes per tissue in our cross-tissue TWAS results (average Spearman's $\rho_{UTMOST} = 0.46$; Supplementary Data 4 and Supplementary Fig. 3). This suggests that cross-tissue TWAS imputation models can effectively select predictive cis eQTL variants (within ±1 Mb of the transcription start/end site of a gene) shared by multiple tissues, whereas tissue-specific eQTLs with strong effects were retained.

### Cross-tissue and multiple-gene-conditioned fine-mapping analyses at loci with multiple-gene-disease association signals

$P_{GBJ\_marginal}$ values from cross-tissue TWAS analyses (Supplementary Data 3) showed weak to moderate inflation of GBJ association statistics ($\lambda_{1000}$ in the range [0.94; 1.05]; Supplementary Data 5). One reason for this may be co-regulation of multiple genes by the same eQTL, as well as LD between eQTLs with nonzero prediction weights, which may lead to correlated predicted expression between nearby genes and thus multiple correlated TWAS hits per locus[15], analogous to the situation in a GWAS study where an association

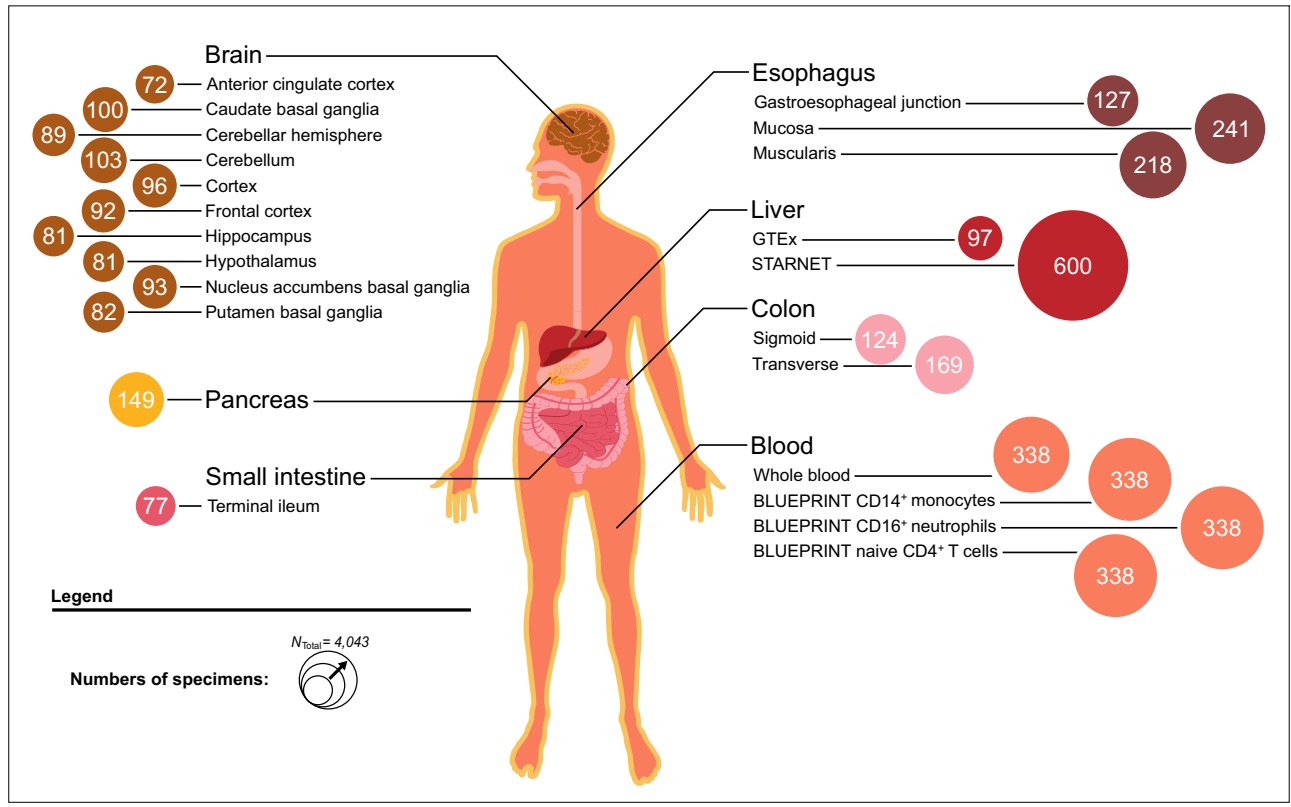

**Fig. 1 Selection of 23 tissue and cell types to perform transcriptome-wide association analyses (TWAS) for gut-brain axis (GBA) diseases.** We selected 23 tissues and cell types previously found to contribute to the heritability of psychiatric or immune-mediated disease (Methods). Cross-tissue TWAS training and imputation models were used using transcriptome-wide and genome-wide reference data from 4043 reference samples from 23 tissues and cell types (Supplementary Data 2) provided by consortium projects GTEx, STARNET, and BLUEPRINT (Methods): gastrointestinal and liver tissues (*n* = 9), brain tissues (*n* = 10), whole blood (*n* = 1), and immune cell types (CD14+ monocytes, CD16+ neutrophils, CD4+ T-cells; *n* = 3). Different colors represent biological samples from different tissue classes. The size of the circles corresponds to the number of reference samples of the respective tissue.

signal typically spans a large number of highly significant SNPs in high LD with the causal SNP[26]. In order to distinguish between causal and marker gene-disease associations at significant TWAS loci, we performed (i) multiple-gene-conditioned single-tissue analysis (Single-tissue$_{conditional}$) followed by (ii) multiple-gene-conditioned cross-tissue analysis (GBJ$_{conditional}$) (Methods) with nearby genes within range of ±1 Mb around transcriptome-wide significant genes from marginal results (genes highlighted as red dots in the Manhattan plots in Supplementary Fig. 2). This accounted for potential cross-tissue co-regulation and LD between eQTL SNPs, and furthermore, the marginal association result of each gene was conditioned based on all genes within the ±1 Mb region (avoiding specification of fixed locus boundaries based on LD blocks from genotype data). This analysis yielded a total of 215 (80), 152 (50), 150 (22), 89 (53), 9 (5), 5 (5), 10 (6) GBJ gene-disease associations (separate loci of ±1 Mb) with transcriptome-wide significance ($P_{GBJ\ conditional} < 3.20 \times 10^{-6}$) for CD, UC, PSC, SCZ, MDD, BD, ADHD, respectively (Supplementary Data 3 and Supplementary Data 6, Methods), thus representing final transcriptome-wide significant cross-tissue multiple-gene-conditioned TWAS gene-disease association results (Table 1 and Fig. 2a) within and outside the boundaries of established GWAS susceptibility loci (Supplementary Data 7).

**Comorbidity analysis and trait correlation at the genetic level and the level of predicted gene expressions.** Using population-based administrative health records from the Danish National Patient

Registry (DNPR), we conducted a retrospective cohort study for the period 1994 to 2018 by examining directed diagnosis pairs of 7,191,385 people from the entire Danish population (Methods). We confirmed statistically significant comorbidity (false discovery $P_{FDR} < 0.05$) between our four psychiatric and three immunological phenotypes for nine psychiatric-immunological pairs out of 42 disease pairs (Supplementary Data 8) with an increased incidence of SCZ (relative risk RR = 1.15), BP (RR = 1.28) and depression (RR = 1.52) in CD, depression in UC (RR = 1.34) and PSC (RR = 1.42), CD in SCZ (RR = 1.12), BD (RR = 1.29) and depression (RR = 1.57), UC (RR = 1.37) and PSC (RR = 1.57) in depression. We further applied pairwise genetic correlation analysis with LDSC as described in Tylee et al.[27] (Methods) to complement TWAS analyses with up-to-date values of genetic correlations (Supplementary Data 9 and Supplementary Fig. 4).

Next, to examine pairwise trait correlations at the level of predicted (genetically regulated) gene expression ($\rho_E$) and to compare $\rho_E$ with genome-wide genetic correlations ($\rho_G$), we calculated Spearman's correlations between TWAS gene effect estimates for all pairs of diseases by using the gene-wise Z-scores from the analyses Single-tissue$_{marginal}$ and GBJ$_{marginal}$ and a correlation-pruned list of 1825 co-regulation- and eQTL-independent genes previously used in Mendelian randomization studies[28] (to avoid measuring correlated predicted expression on the transcriptome-wide scale) (Methods). As an example (Single-tissue$_{marginal}$; Supplementary Data 10), the strongest positive correlation across psychiatric and immune phenotypes was observed for CD4+ T-cells ($\hat{\rho}_E = 0.11$ for CD and BP and

$\hat{\rho}_E = 0.14$ for CD and SCZ) (Supplementary Fig. 5). On the cross-tissue level across all 23 tissues (GBJ$_{marginal}$; Supplementary Data 10), the strongest positive correlations across psychiatric and immune phenotypes were observed for pairs UC and BP ($\hat{\rho}_E = 0.15$), CD and BP ($\hat{\rho}_E = 0.14$), UC and SCZ ($\hat{\rho}_E = 0.13$), and CD and SCZ ($\hat{\rho}_E = 0.11$) (Fig. 2b). This differs slightly from the genetic level for pairs UC and BP ($\hat{\rho}_G = 0.07$), CD and BP ($\hat{\rho}_G = 0.11$), but may explain part of the genome-wide genetic correlations ($\hat{\rho}_G$) of UC and SCZ ($\hat{\rho}_G = 0.13$), and CD and SCZ ($\hat{\rho}_G = 0.11$), suggesting that part of the effects of genetic variants shared between SCZ and immune phenotypes are likely to be mediated by gene expression.

**Mendelian randomization analysis for psychiatric and immune phenotypes.** Mendelian randomization (MR) is a commonly used method to assess a causal influence of one trait (exposure) on another (outcome). To identify putative pairs of immune-mediated and psychiatric diseases that may be related in a causal manner (i.e., vertical pleiotropy[29]), we conducted MR analysis for transcriptome-wide significant lead genes (i.e., our instruments with $P_{conditional} < 3.20 \times 10^{-6}$ and filtered for the most-significant genes from Single-tissue$_{conditional}$ within $\pm 1$ Mb regions) using multi-gene TWAS MR analysis (including HEIDI-outlier filtering that removes instruments with strong putative horizontal pleiotropic effects, where one gene has independent effects on multiple traits) as implemented in GSMR[30] (Methods). After Bonferroni correction for 171 pairwise tests ($P_{GSMR} < 2.92 \times 10^{-4}$), we identified CD ($P_{GSMR} = 9.58 \times 10^{-5}$; OR = 1.10, 95% CI 1.05–1.15) and UC ($P_{GSMR} = 8.40 \times 10^{-6}$; OR = 1.17, 95% CI 1.09–1.26) as a putative risk factor (exposure) for SCZ in tissues "Colon_-Transverse" and "Esophagus_Gastroesophageal_Junction", respectively (Supplementary Data 11). HEIDI-filtering identified *CSNK1A1* and *SGSM3* as potential horizontally pleiotropic genes, suggesting a pleiotropic function in both diseases.

**Identification of gene-disease associations shared between psychiatric and immune phenotypes.** After identifying pairs of immune-mediated and psychiatric diseases as putative correlated traits on the transcriptome-wide level, we sought to identify susceptibility genes shared between psychiatric and immune phenotypes from the list of all transcriptome-wide significant genes from both analyses Single-tissue$_{conditional}$ and GBJ$_{conditional}$ (Table 1). Out of 288 possible pairwise tissue-specific combinations of psychiatric and immune phenotypes (Supplementary Data 12), gene overlap analysis revealed 31 (of those 4 GBJ$_{conditional}$) pairs of psychiatric and immune phenotypes that share at least one susceptibility gene. After excluding genes within the major histocompatibility complex (MHC; chr6: 25–34 Mb), which was particularly relevant to pairs PSC-MDD and PSC-SCZ in various tissues, we identified three non-MHC TWAS susceptibility genes (*NR5A2*, *SATB2*, *PPP3CA*) from analyses Single-tissue$_{conditional}$ to be shared between brain (SCZ, CD, UC: Hypothalamus; SCZ, UC: Frontal cortex BA9; SCZ, CD: Putamen basal ganglia) and intestinal tissues (CD, UC: Colon transversum; UC: Colon sigmoideum; CD: Colon transversum) (Table 2; more detailed results in Supplementary Data 3). In summary, these three genes met the transcriptome-wide significance threshold of $3.20 \times 10^{-6}$ in the same brain or non-brain tissue for at least one immune disease and one psychiatric disease, where at least one of the diseases must also show another transcriptome-wide significant signal for another brain or non-brain tissue (brain tissue if the first signal occurred in a non-brain tissue or vice versa), with substantial gene expression heritability estimates $h_{med}^2$ (i.e., heritability mediated by the *cis* genetic component of gene expression levels) across the 23 tissues ($h_{med-min}^2$, $h_{med-max}^2$) of (13.4%, 18.2%) for *NR5A2*,

**Table 1 Transcriptome-wide significant ($P_{conditional} < 3.20 \times 10^{-6}$) genes from multiple-gene-conditioned fine-mapping analysis at loci with multiple-gene-disease association signals, for psychiatric and immune phenotypes in 23 gut-brain-axis (GBA) tissues.**

| | CD | UC | PSC | SCZ | MDD | BD | ADHD |
|---|---|---|---|---|---|---|---|
| TWAS discovery $n_{cases}/n_{controls}$ across 23 tissues | 18,431/33,658 | 14,191/33,658 | 3,377/33,658 | 36,989/113,075 | 135,458/344,901 | 9412/127,760 | 20,183/35,191 |
| n unique genes [loci] across 23 tissues | 215 [80] | 152 [50] | 150 [22] | 89 [53] | 9 [5] | 5 [5] | 10 [6] |
| n unique genes in total from single tissues | 242 (240) | 171 (163) | 136 (61) | 108 (85) | 9 (5) | 14 (14) | 11 (11) |
| n unique genes GWAS loci | 164 (164) | 109 (109) | 51 (50) | 52 (49) | 2 (2) | 7 (7) | 3 (3) |
| n unique genes non-GWAS loci | 78 (76) | 62 (54) | 85 (11) | 56 (36) | 7 (3) | 7 (7) | 8 (8) |
| n unique genes cross-tissue* | 132 (132) | 89 (88) | 81 (41) | 56 (42) | 4 (3) | 3 (3) | 5 (5) |
| n unique genes cross-tissue** brain and non-brain | 99 (99) | 67 (67) | 57 (32) | 38 (28) | 3 (2) | 2 (2) | 4 (4) |

For the selection of 23 tissues, see Fig. 1, Supplementary Data 2 and Methods. 164 (78), 109 (62), 51 (85), 52 (56), 2 (7), 7 (7), 3 (8) TWAS susceptibility genes are within (outside) the boundaries of established GWAS susceptibility loci (Methods, Supplementary Data 7) for CD, UC, PSC, SCZ, MDD, BD, and ADHD, respectively. Approximately half of all gene-disease associations are found in at least two tissues of the GBA (for more details see Supplementary Data 6). TWAS discovery $n_{cases}/n_{controls}$: number of cases and controls of GWAS data sets (Supplementary Data 1); $n$ unique genes [loci] across 23 tissues, number of transcriptome-wide significant genes ($P_{GBJ\,conditional} < 3.20 \times 10^{-6}$, Fig. 2a) from cross-tissue and multiple-gene-conditioned fine-mapping analysis at loci with multiple-gene-disease association signals (analysis GBJ$_{conditional}$; see Methods). The number of associated loci of size $\pm 1$ Mb in square brackets corresponds to the number of red dots in Fig. 2a. GBJ$_{conditional}$ corrected for nearby genes in the range of $\pm 1$ Mb around for leading transcriptome-wide significant genes ($P_{conditional} < 3.20 \times 10^{-6}$; Supplementary Data 3) from multiple-gene-conditioned single-tissue analysis (i.e., Single-tissue$_{conditional}$, before applying the generalized Berk-Jones (GBJ) test to quantify total gene-disease associations across all 23 tissues).

*CD* Crohn's disease, *UC* ulcerative colitis, *PSC* primary sclerosing cholangitis, *SCZ* schizophrenia, *MDD* major depressive disorder, *BIP* bipolar disease (BIP), *ADHD* attention-deficit/hyperactivity disorder.

*significant gene-disease associations in at least two tissues.

**significant gene-disease associations in at least two tissues including the brain and non-brain tissues (excluding cross-tissue associations for immune and gastro/liver tissues only, respectively, see Fig. 1); in curved brackets: number of gene-disease associations outside the extended major histocompatibility complex (MHC; chr6:25–34 Mb).

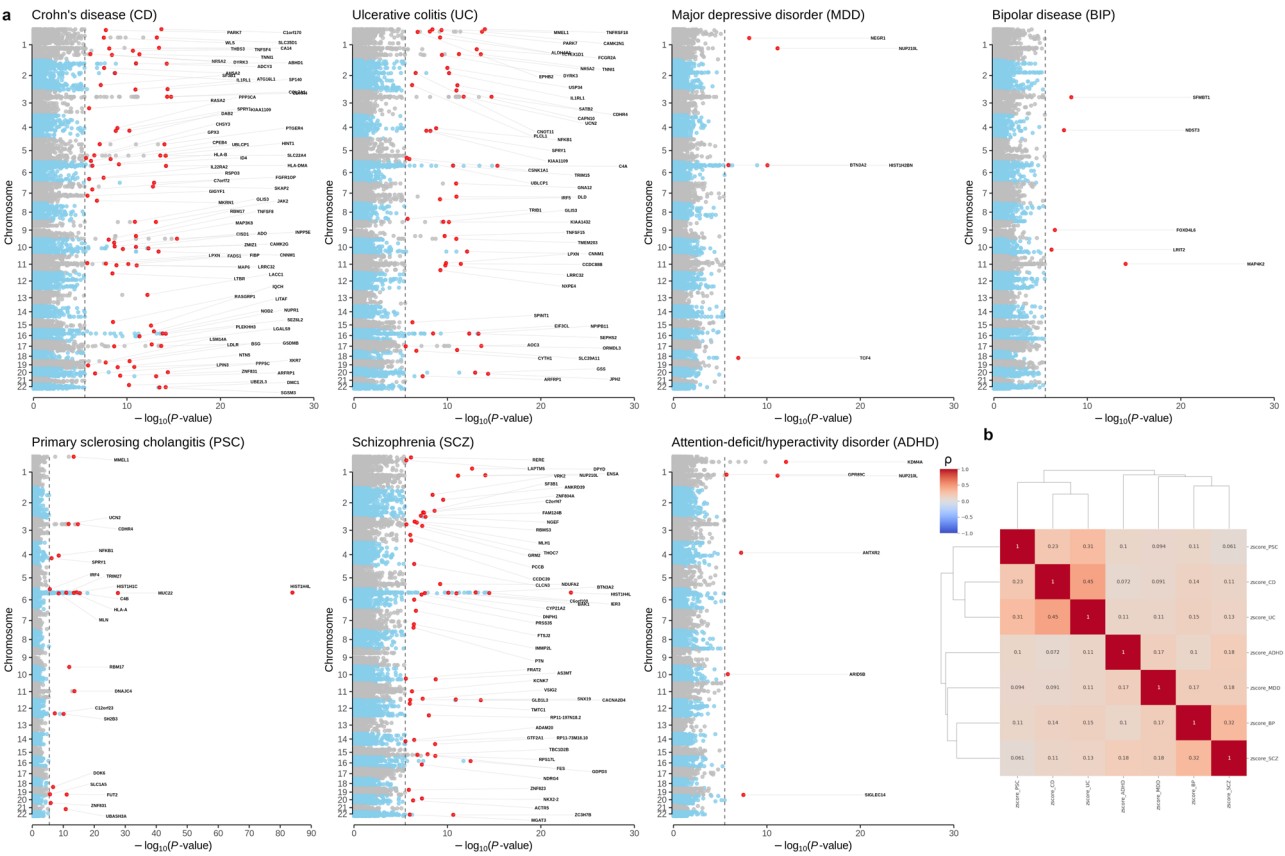

**Fig. 2 Manhattan plots showing transcriptome-wide significant ($P_{conditional} < 3.20 \times 10^{-6}$ from $GBJ_{conditional}$) gene discoveries for psychiatric and immune phenotypes in 23 gut-brain-axis (GBA) tissues, and proportion of shared predicted transcriptome-wide disease-associated gene expression ($\rho_E$) for pairs of immune-mediated and psychiatric diseases averaged across 23 tissues of the GBA. a** Gray dotted line, transcriptome-wide significance threshold; red dots, lead gene-disease associations at TWAS loci (for regions of size ±1 Mb) and conditioned on all genes within ±1 Mb regions. CD Crohn's disease, UC ulcerative colitis, PSC primary sclerosing cholangitis, SCZ schizophrenia, MDD major depressive disorder, BIP bipolar disease, ADHD attention-deficit/hyperactivity disorder. **b** Spearman's trait correlation (ρ) values at the transcriptome-wide level of gene expression for each disease pair and averaged across 23 tissues of the GBA show that effects of genetic risk variants shared between psychiatric and immune phenotypes are largely mediated by gene expression. Strongest positive correlations across psychiatric and immune phenotypes were observed for pairs UC and BP ($\hat{\rho}_E = 0.15$), CD and BP ($\hat{\rho}_E = 0.14$), UC and SCZ ($\hat{\rho}_E = 0.13$), and CD and SCZ ($\hat{\rho}_E = 0.11$), which is in line with genome-wide genetic correlations ($\hat{\rho}_G$) for pairs UC and BP ($\hat{\rho}_G = 0.23$), CD and BP ($\hat{\rho}_G = 0.22$), UC and SCZ ($\hat{\rho}_G = 0.14$), and CD and SCZ ($\hat{\rho}_G = 0.12$) reported by Tylee et al.[11]. The dendrogram reflects the corresponding hierarchical single linkage clustering across diseases. Genetic trait correlations for single-tissue results (analysis Single-tissue$_{conditional}$) are given in Supplementary Data 9, 10. For the selection of 23 tissues, see Fig. 1, Supplementary Data 2, and Methods. Results from unconditioned cross-tissue TWAS analysis ($GBJ_{marginal}$; before multiple-gene-conditioned fine-mapping analysis) are shown in Supplementary Fig. 2.

(6.9%, 24.8%) for *SATB2*, and (5.0%, 10.9%) for *PPP3CA* (Table 2; Methods). Thus, these genes met the most stringent criteria for shared susceptibility genes, as they were shared between psychiatric and immune phenotypes in the same tissues and across the brain and non-brain tissues and were therefore prioritized below. We observed that an increase (decrease) in predicted expression for *NR5A2* and *SATB2* (*PPP3CA*) was associated with an increased risk of SCZ, CD, and/or UC (Table 2). In addition, four other gene candidates (*INO80E, SF3B1, SGSM3,* and *ZC3H7B*) from the secondary analysis ($GBJ_{conditional}$) met the transcriptome-wide significance threshold and showed associations across both brain and intestinal tissues, but each in different tissues for psychiatric and immune phenotypes, implying that for these four genes the most significant tissues differ between psychiatric and immune phenotypes (Supplementary Data 12–14). This suggests that those four genes may be involved in SCZ and CD/UC, but not necessarily in the same tissue. We will show below by gene set enrichment analysis that *INO80E, SGSM3,* and *ZC3H7B* are likely to be proxies for other gene-disease associations at the respective loci.

**GWAS/TWAS conditional analysis to describe the GWAS contribution to the TWAS result**. To estimate the contribution of the three shared same-tissue susceptibility genes *NR5A2, SATB2,* and *PPP3CA* relative to the established GWAS signals as defined by the recent consortia fine-mapping studies for SCZ[31] and CD/UC[32,33], we performed a joint GWAS/TWAS fine-mapping analysis across the 23 tissues of the GBA using the GWAS consortium summary statistics listed in Supplementary Data 1. For *NR5A2, SATB2,* and *PPP3CA*, respectively, we performed a multi-SNP-based conditional and joint association analysis using GCTA COJO[34] to condition GWAS summary statistics on independent eQTL effects of the three genes (Methods). We examined how much of the GWAS fine-mapping signal remained after the signal of the TWAS gene eQTLs was removed. We demonstrated the applicability of our approach by examining and replicating established eQTL associations of GWAS signals as reported in the fine-mapping GWAS studies for CD/UC (Supplementary Note 1 and Supplementary Figs. 6, 7; detailed results showing all genes from the regions around *NR5A2, SATB2, PPP3CA, INO80E SF3B1, SGSM3,* and *ZC3H7B*

**Table 2 Transcriptome-wide significant susceptibility genes (NR5A2, SATB2, PPP3CA) shared between schizophrenia and inflammatory bowel diseases expressed in the gut-brain axis (GBA) tissues.**

| Gene | Gene-start-end (in kb) | ($h^2_{med-min}$, $h^2_{med-max}$) in % | Primary tissue(s) of strongest gene-disease association | Diseases primary tissue(s) | Secondary tissue(s) of gene-disease association | $Z_{marginal}$ primary tissue | | | $P_{conditional}$ primary tissue | | | Overlap with GWAS locus | Distance to GWAS locus |
|---|---|---|---|---|---|---|---|---|---|---|---|---|---|
| | | | | | | SCZ | CD | UC | SCZ | CD | UC | | |
| NR5A2 | chr1:199,996 - 200,146 | (13.4, 18.2) | Hypothalamus | SCZ, CD, UC | SCZ-; CD/UC: Colon transversum | 4.86 | 5.82 | 7.7 | $1.63 \times 10^{-6}$ | $3.66 \times 10^{-8}$ | $1.97 \times 10^{-13}$ | No overlap | SCZ:104 kb; CD/UC:654 kb |
| SATB2 | chr2:200,134 - 200,320 | (6.9, 24.8) | Frontal Cortex BA9 | SCZ, UC | SCZ -; UC: Colon sigmoideum | 4.75 | -0.55 | 8.46 | $1.29 \times 10^{-6}$ | 0.908 | $2.27 \times 10^{-13}$ | No overlap | SCZ:67 kb; UC:425 kb |
| PPP3CA | chr4:101,947 - 102,267 | (5.0, 10.9) | Putamen basal ganglia | SCZ, CD | SCZ: - CD: Colon transversum | -4.87 | -6.79 | -2.96 | $1.57 \times 10^{-6}$ | $1.28 \times 10^{-10}$ | $5.54 \times 10^{-3}$ | No overlap | SCZ:280 kb; CD:384 kb |

Gene overlap analysis of TWAS results from multiple-gene-conditioned fine-mapping analysis (Single-tissue$_{conditional}$; Supplementary Data 3) identified three common TWAS susceptibility genes (NR5A2, PPP3CA, SATB2) that meet the transcriptome-wide significance threshold ($P_{conditional}$ < 3.20 × 10⁻⁶ for each disease separately; for diseases listed in the "Diseases primary tissue(s)" column). These three genes are shared between psychiatric and immune phenotypes in the same tissues and across the brain and non-brain tissues (Fig. 3). Increased predicted expression (indicated by a positive Z-score) of NR5A2 and SATB2 is associated with increased risk SCZ and CD as well as SCZ and UC. Decreased predicted expression (indicated by a negative Z-score) of PPP3CA is associated with increased risk for SCZ and CD. Genes inside the extended major histocompatibility complex (MHC, chr6:25–34 Mb) were excluded from gene overlap analysis. Gene, susceptibility gene shared between psychiatric and immune disease(s); Gene-start-end, transcription start-end positions (including UTRs) in kilobases (genome build GRCh37/hg19); ($h_{med-min}^2$, $h_{med-max}^2$), estimated minimum/maximum gene expression heritability in percentage attributable to SNPs in the vicinity of each gene (Methods) and averaged across 23 different tissues (Fig. 1); Primary tissue(s) identified from multiple-gene-conditioned fine-mapping analysis (Single-tissue$_{conditional}$, Methods), in which the transcriptome-wide significant gene-disease association signal occurred; Diseases primary tissue(s), diseases showing a transcriptome-wide association signal in the primary tissue(s) in Single-tissue$_{conditional}$ analysis; CD, Crohn's disease; UC, ulcerative colitis; SCZ, schizophrenia; Secondary tissue(s), further tissues with suggestive evidence ($P_{conditional}$ < 1 × 10⁻⁴) for gene-disease association; Z statistic, Z-score from analysis Single-tissue$_{conditional}$ and/or GBJ$_{conditional}$ for primary tissue(s), the plus/minus sign indicates increased/ decreased predicted expression of these genes to be associated with increased disease risk, TWAS $P_{conditional}$, P value from multiple-gene-conditioned fine-mapping analyses Single-tissue$_{conditional}$ and/or GBJ$_{conditional}$ for primary tissue(s); P values' significance threshold was 3.20 × 10⁻⁶; Overlap with GWAS locus, testing an overlap with locus boundaries of established GWAS loci from the latest fine-mapping GWAS studies for SCZ[31] and CD/UC[32,33]. Distance to GWAS locus, closest GWAS loci from literature for SCZ[31], CD, and UC[33] within ± 1 Mb region around coordinates from column "Gene-start-end" (in kb; genome build GRCh37/hg19): SCZ: chr1:200,250-200,422 (1q32.1) with candidate gene LINC00862, ZNF281. CD/UC: chr1:200,803-201,092 (1q32.1) with candidate genes CAMSAP2, RPL34P6, MRO3HP3, KIF21B, GPR25, CACNA1S, C1orf106, ASCL5. UC: chr2:199,447-199,709 (2q33.1). SCZ: chr2:200,387-200,633 (2q33.1) with candidate genes FTCDNL1, LOC101927641. SCZ: chr2:200,536-201,310 (2q33.1) with candidate genes C2orf47, C2orf69, FTCDNL1, SPATS2L, TYW5. SCZ: chr4:102,547-103,389 (4q24) with candidate genes BANK1, SLC39A8. CD: chr4:102,651-103,144 (4q24) with candidate gene BANK1.

and all significant tissues are described in the Supplementary Note 2 and Supplementary Figs. 8–37). Joint GWAS/TWAS fine-mapping results for the additional gene candidates (INO80E, SF3B1, SGSM3, and ZC3H7B) from analyses GBJ$_{conditional}$ are discussed in Supplementary Fig. 38–41 and Supplementary Note 4.

Increased genetically regulated expression of NR5A2 in the hypothalamus and colon transversum was associated with increased risk for SCZ, CD, and UC. We found that the TWAS association of NR5A2 at 1q32.1 in the hypothalamus and colon transversum (Fig. 3a) was driven by the nearby (but nonoverlapping) established SCZ and CD/UC GWAS signals located 104 and 654 kilobases (kb) upstream of NR5A2, respectively. This GWAS locus was not assigned a gene in the CD/UC consortium fine-mapping study[33] but was assigned an overlap with an epigenetic mark for immune cells (Immune_H3K4me1). Given the functional evidence for NR5A2 in attenuating inflammatory damage in murine and human intestinal organoids[35] (see also Supplementary Note 3) we propose NR5A2 as the best candidate gene for the GWAS locus 1q32.1 with a possible effect on the hypothalamic–pituitary–adrenal axis (HPA axis)[10].

Increased genetically regulated expression of SATB2 in frontal cortex BA9 and colon sigmoideum was associated with increased risk of SCZ and UC. The TWAS association of SATB2 at 2q33.1 (Fig. 3b) is driven by the nearby (nonoverlapping) established SCZ and UC GWAS signals located 67 and 425 kilobases (kb) downstream and upstream of SATB2, respectively. Because multiple TWAS eQTLs were identified for SATB2 in the 95% credible set of the UC GWAS peak (consisting of intergenic variants between LOC101927619 and SATB2[33]), it is convincing that the GWAS/TWAS fine-mapping analysis revealed a possible explanation of the GWAS signal by eQTLs at this locus. SATB2 as a potential susceptibility gene for UC and SCZ has already been implicated by other studies (Supplementary Note 3).

Decreased genetically regulated expression of PPP3CA in the putamen basal ganglia and colon transversum was associated with increased risk of SCZ and CD. We found that the TWAS association of PPP3CA at 4q24 (Fig. 3c) is driven by the nearby (nonoverlapping) established SCZ and CD GWAS signals located 280 and 384 kilobases (kb) upstream of PPP3CA, respectively. GWAS fine-mapping analysis[33] assigned the 95% credible set of fine-mapped SNP variants ($n$ = 170 SNPs) to the gene BANK1, which is closest to the lead GWAS SNP variant at 4q24. Because PPP3CA encodes a subunit of calcineurin (Supplementary Note 3) that has been associated with SCZ and is also a known drug target (due to a direct physical interaction with FKBP Prolyl Isomerase 1 A (FKBP1A)[36], also known as FKBP12) for the treatment of UC via calcineurin inhibitors, we propose PPP3CA as the best candidate gene for the GWAS association signal at 4q24.

**Analysis of bulk RNA-Seq data of different developmental stages and scRNA-seq expression data for tissues of the GBA.** Having tested the three gene-disease associations for plausibility at the GWAS level in the previous section, we hypothesized that high, tissue- or cell-type-specific expression would be indicative of a particular function in the tissue of interest, and that expression of a gene could also occur only in a small subset of cells of only a particular tissue or at a particular human developmental stage, which is hypothesized for SCZ[37]. We used publicly available non-diseased bulk tissue[38,39] and scRNA-Seq data sets[40,41] (Methods) to look up in which tissues and at which developmental stage of human tissues and cells the shared susceptibility genes are generally expressed, and found that the observed patterns matched the existing knowledge on the genes (Supplementary Note 3). It is also possible that an adverse event

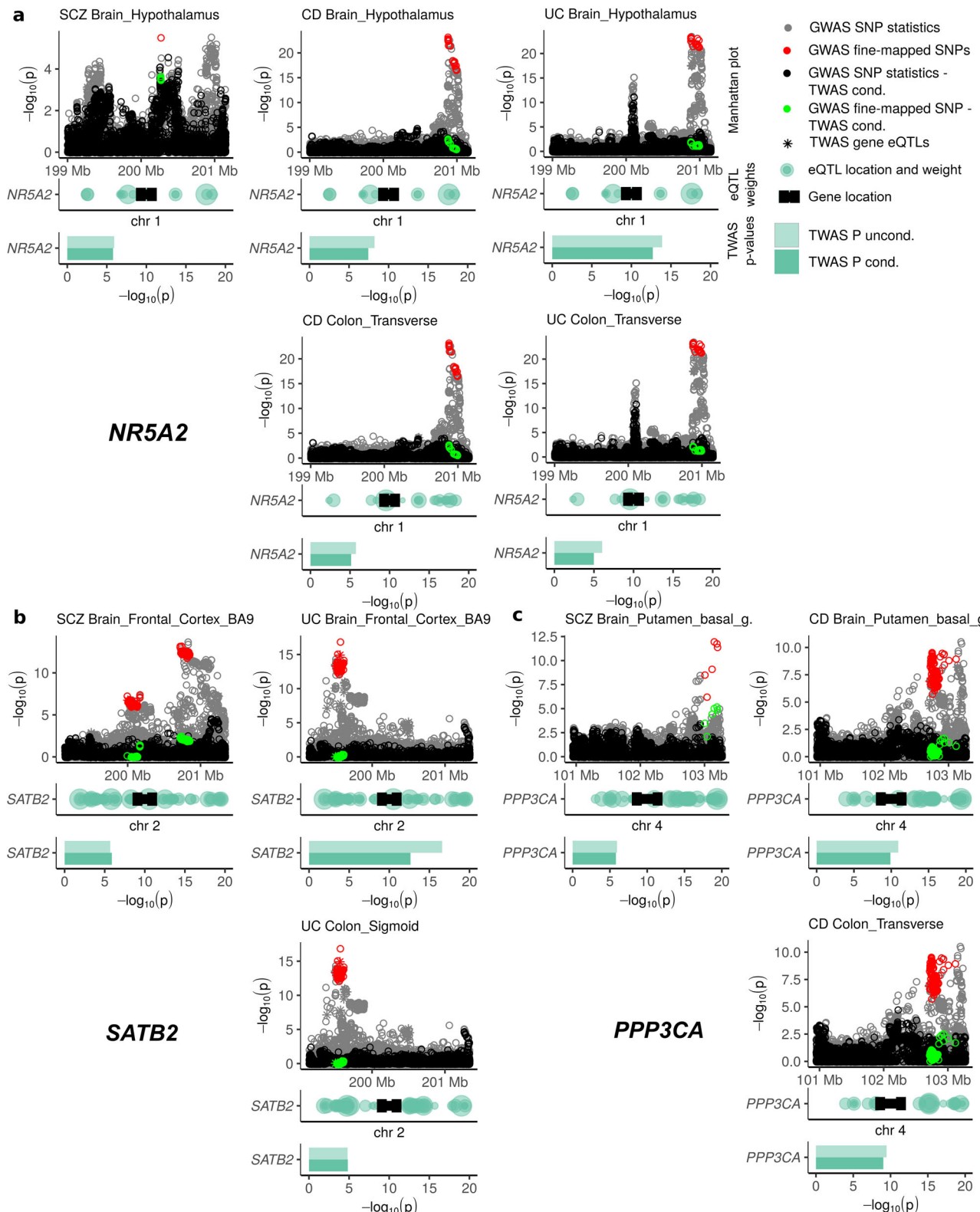

in utero can interfere with normal brain development and create a brain vulnerability that, in an individual already at risk (genetic predisposition), can lead to SCZ later in life[37]. We, therefore, examined the expression of our three candidate genes in developmental data from brain[38] and intestine[39], since it is conceivable that a susceptibility gene is expressed during fetal development but is no longer expressed in adulthood. Again, *NR5A2* was almost absent in bulk brain samples from different developmental stages, whereas *SATB2* was strongly upregulated specifically in the fetal forebrain during midfetal development, and *PPP3CA* was weakly expressed in all developmental stages of the brain[38] (Fig. 4a). In the intestine, it was the opposite: *NR5A2* was strongly expressed in the ileum and colon during the fetal and juvenile stages, whereas *SATB2* was weakly expressed in the colon sigmoideum and not ileum, and *PPP3CA* almost not at all[39] (Fig. 4a). To achieve resolution at the single-cell level, we

**Fig. 3 The transcriptome-wide significant gene-disease association signals of *NR5A2* (1q32.1), *SATB2* (2q33.1), and *PPP3CA* (4q24) for SCZ, CD, and UC largely explain fine-mapped GWAS association signals from nearby GWAS susceptibility loci. a** The gene-disease association signal of *NR5A2* is driven by the nearby but nonoverlapping established SCZ and CD/UC GWAS signals located 104 and 654 kilobases (kb) upstream of *NR5A2*, respectively (Table 2). Increased genetically regulated expression of *NR5A2* in the hypothalamus and colon transversum was associated with increased risk for schizophrenia (SCZ), Crohn's disease (CD), and ulcerative colitis (UC). **b** The association signal of *SATB2* is driven by the nearby but nonoverlapping established SCZ and UC GWAS signals located 67 and 424 kilobases (kb) downstream and upstream of *SATB2*, respectively. Increased genetically regulated expression of *SATB2* in frontal cortex BA9 and colon sigmoideum is associated with increased risk for SCZ and UC. Although 95% credible sets of variants most likely to be causal (red circles; upper part) differ between SCZ and UC, the distribution pattern of TWAS gene eQTLs of *SATB2* (gray asterisks [upper part] and green transparent circles [middle part]) leads to the shared gene-disease association for *SATB2* in both SCZ and UC. **c** The gene-disease association signal of *PPP3CA* is driven by the nearby (nonoverlapping) established SCZ and CD GWAS signals located 280 and 383 kilobases (kb) upstream of *SATB2*, respectively. Decreased genetically regulated expression of *PPP3CA* in putamen basal ganglia and colon transversum is associated with increased risk for SCZ and CD. Raw data of this figure can be found in Supplementary Data 16. Explanation of figure elements: Subplots include original consortium GWAS SNP summary statistics of size ±1 Mb (gray circles; upper part) around the gene, with the 95% credible sets of variants most likely to be causal at each locus (red circles; upper part) as defined by GWAS fine-mapping studies[31,33]. SNP summary stats were conditioned (black circles; upper part) on TWAS gene eQTLs (black asterisks [upper part] correspond to the green transparent circles [middle part], with the relative absolute weight of the eQTLs visualized by the size of green transparent circles) to examine if the original GWAS SNP statistics (gray circles [upper part]; gray asterisks correspond to the green transparent circles [middle part]) can be explained by genetically regulated expression of the gene (green rectangles depict unconditioned/conditioned TWAS *P* values from Single-tissue$_{conditional}$ analysis; lower part). GWAS SNP summary statistics of 95% credible sets of variants after joint GWAS/TWAS fine-mapping analysis are depicted as light green circles (upper part). Gene-start and gene end positions are marked by dumbbell-shaped black bars (middle part).

analyzed human single-nucleus and scRNA-seq data from different cell types of the adult brain[40] and gastrointestinal tract[41], respectively (Fig. 4b). In these data sets, *NR5A2* is expressed in the ileal progenitor, stem, and transient amplifying (TA) cells, less strongly in the colon and rectum, and not in the brain, consistent with the bulk expression data. Consistent with the bulk expression from developmental stages of the intestine (Fig. 4a), *SATB2* was strongly expressed in cell types of the colon and rectum and barely expressed in brain cells. *PPP3CA* expression was highly enriched in enteroendocrine and Paneth-like cells of the ileum, colon, and rectum despite weak expression in the bulk RNA-Seq data of ileum or colon, which may imply a specific role of *PPP3CA* in immune modulation in the gut for CD and UC; in the brain, *PPP3CA* is also expressed in neurons (Fig. 4b). Bulk tissue and scRNA-seq results for genes *INO80E SF3B1*, *SGSM3*, and *ZC3H7B* are shown in Supplementary Fig. 42 and discussed in Supplementary Note 5.

**Calcineurin-dependent NFAT and Wnt signaling as shared signaling pathways for IBD and SCZ.** Based on the statistically unambiguous gene prioritization from multiple-gene-conditioned fine-mapping analyses, it is likely that *NR5A2*, *SATB2*, and *PPP3CA* are shared susceptibility genes for both IBD and SCZ. The genes *INO80E*, *SGSM3*, and *ZC3H7B* from the secondary and not so stringent gene overlap analysis (GBJ$_{conditional}$) might be proxies for other genes with strongly correlating gene expression at the same locus (except for *SF3B1* (Supplementary Figs. 38–42), where the gene-disease association signal of the eleven nearby genes of *SF3B1* clearly disappeared after conditioning, see Supplementary Figs. 8–37), because these three loci showed an extremely high gene density with 45 (*INO80E*), 16 (*SGSM3*), and 29 (*ZC3H7B*) genes (Supplementary Figs 8–37) making the multiple-gene-conditioned fine-mapping analysis difficult[15]. We assumed that these genes, which according to previous findings have nothing to do with IBD or SCZ (Supplementary Note 4), may be masking the causative gene of the locus. To test whether any of the secondary gene candidates could be linked to *NR5A2*, *SATB2*, and *PPP3CA*, we performed gene set enrichment analysis as implemented in EnrichR[42] using the NCATS BioPlanet pathway resource of 1658 curated human pathways[43] and 27 input genes including our seven candidate genes (*NR5A2*, *SATB2*, *PPP3CA*, *SF3B1*, *INO80E*, *SGSM3*, and *ZC3H7B*; Supplementary Data 13), 19 correlated transcriptome-wide significant candidate

genes after multiple-gene conditioned analysis for *INO80E*, *SGSM3* and *ZC3H7B*, and *CSNK1A1* (a pleiotropic HEIDI gene outlier for SCZ and UC from MR analysis (Supplementary Data 11) representing a case of horizontal pleiotropy). The strongest associated term was "calcineurin-dependent NFAT signaling role in lymphocytes" (enrichment $P = 8.57 \times 10^{-7}$; Supplementary Data 15) including genes *CSNK1A1*, *EP300* (same locus as *SGSM3*), and *MAPK3* (same locus as *INO80E*) in addition to our main candidate gene *PPP3CA*. In our view, *EP300* and *MAPK3* are the more likely candidates instead of *SGSM3* and *INO80E*, respectively. Both genes are important signal transducers and are known to be part of or intersect with the Wnt pathway (enrichment $P = 1.23 \times 10^{-3}$; Supplementary Data 15), which activates NFAT via calcineurin signaling[43,44]. Interestingly, LRH-1, the gene product of *NR5A2* (Supplementary Note 3), has been shown to bind to β-catenin[45], which is also part of the Wnt pathway. Recent findings also suggest that SCZ and BD are characterized by abnormal Wnt gene expression and plasma protein levels, suggesting that suggest that drugs targeting the Wnt signaling pathway may play a role in the treatment of severe mental disorders[46]. Thus, the Wnt signaling pathway and NFAT activation are the most likely candidate pathways for IBD and SCZ, where risk is mediated by gene expression.

## Discussion

By performing cross-tissue and multiple-gene conditioned transcriptome-wide association studies (TWAS) for 23 tissues of the GBA for three clinically related inflammatory diseases of the gastrointestinal tract (CD, UC, and PSC) and four diseases of the mind and brain (SCZ, MDD, BD, and ADHD) we identified susceptibility genes *NR5A2*, *SATB2*, and *PPP3CA* shared between IBD (CD/UC) and SCZ that are (genetically) expressed both in intestinal and brain tissues. We estimated the proportion of the genetically regulated component of expression ($h_{med-min}^2$, $h_{med-max}^2$) with respect to total gene expression, measured across a maximum of 23 different tissues, to be (13.4%, 18.2%) for *NR5A2*, (6.9%, 24.8%) for *SATB2*, and (5.0%, 10.9%) for *PPP3CA*. These three genes could represent a molecular link between psychiatric and inflammatory traits and may contribute to the observed genetic correlations[11,47] and increased prevalence of SCZ in CD/UC patients at the population level[7–9]. The correlations between BD and CD/UC are enigmatic. Tylee et al. find BD and CD/UC correlated on the genetic level, but they cite a study in which BD-UC was not significant and we find

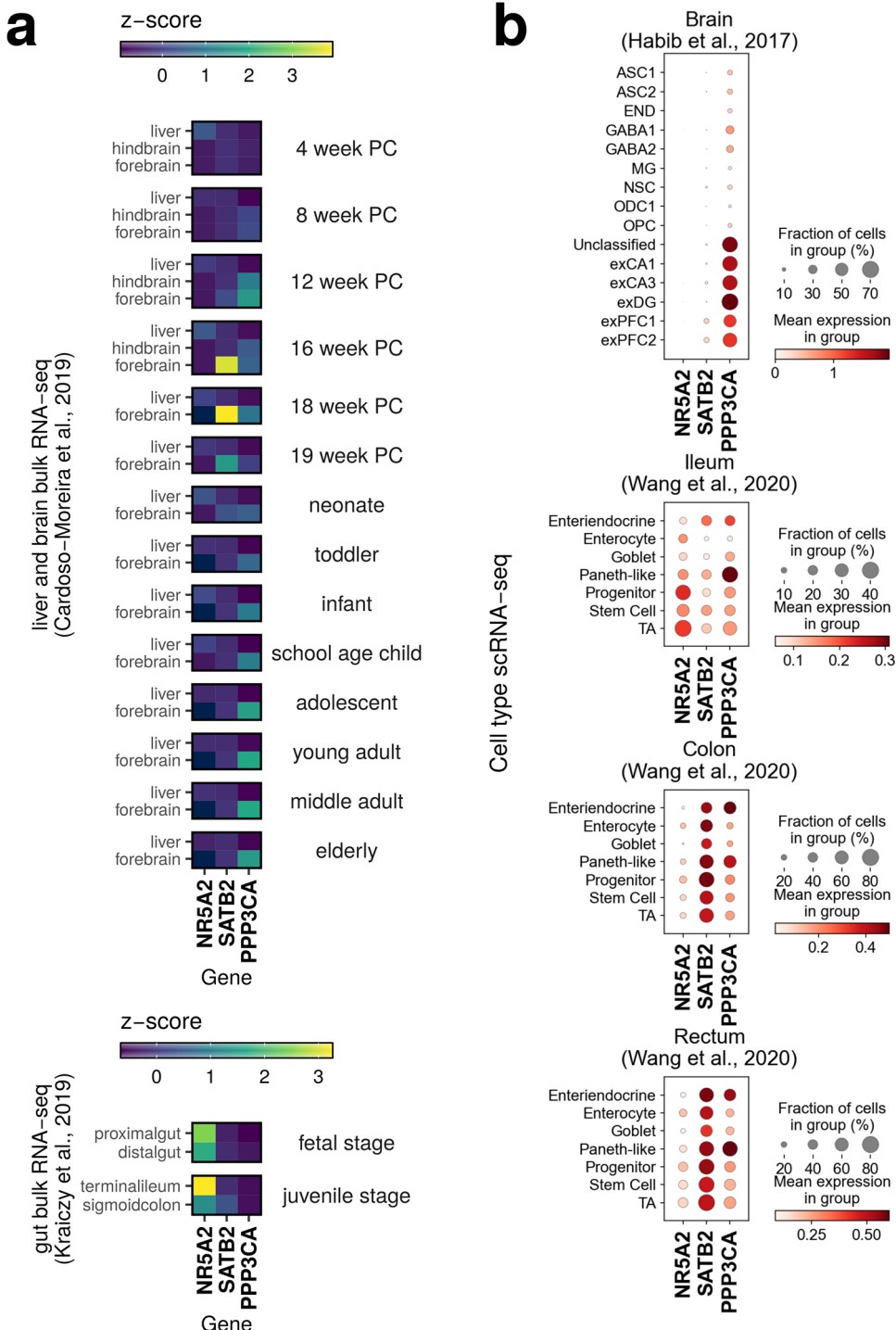

no significance, too. Contrarily, we find BD and CD/UC correlated on the genetic expression level, but no single gene overlap hits. A possible explanation is that Tylee et al used a different dataset for BD. BD is closely related to SCZ and depending on the inclusion criteria, might appear more or less similar to SCZ.

Our bioinformatic cross-tissue TWAS pipeline for selected tissues of GBA addressed typical technical drawbacks of previous TWAS studies and enabled conditional analysis of multiple disease associations at TWAS loci. In our opinion, TWAS analyses for CD, UC, and SCZ had the highest statistical power to detect shared TWAS susceptibility genes, as specifically for CD, UC, and SCZ a much higher proportion of expression-mediated heritability ($h_{\mathrm{med}}^2/h_g^2$) has recently been estimated across GTEx tissues than

for other psychiatric disorders such as BD[48], so our TWAS analyses for BD had lower statistical power. Since $h_{\mathrm{med}}^2/h_g^2$ can vary considerably for different traits or diseases, we recommend that future TWAS studies additionally estimate $h_{\mathrm{med}}^2/h_g^2$ (e.g., with the MESC software[48]) to assess whether a TWAS can be expected to have a low or rather high number of gene-disease associations for the disease under study. Another key outcome of our GWAS/ TWAS conditional analysis for these three genes is that the expression-disease associations largely explain the previously fine-mapped GWAS association signals at nearby GWAS susceptibility loci for SCZ[31], CD, and UC[32,33], suggesting that risk is mediated by genetically regulated expression at those loci. Together with the results of our gene set enrichment pathway analysis, we showed it

**Fig. 4 Analysis of developmental and single-cell RNA-seq data suggest a cell-specific pleiotropic role of *NR5A2*, *SATB2*, and *PPP3CA* in SCZ, CD, and UC and showed that *PPP3CA* expression was strongest in neurons and enteroendocrine and Paneth-like cells of the ileum, colon, and rectum, indicating a possible link to the GBA.** Visualizations of bulk expression of the candidate genes can be found in Supplementary Fig. 42. **a** Brain expression data from humans of different ages[38] confirm GTeX and BLUEPRINT tissue data and show that *NR5A2* is almost absent in brain samples. *SATB2* is strongly upregulated in the fetal forebrain in midfetal development. This suggests a developmental function that may predispose to disease when dysregulated. *PPP3CA* expression is strongest in the forebrain and increases with age. Expression in the liver is given for reference. PC, post conception. Data on expression in the developmental gut[39] show that *NR5A2* is strongly expressed in the ileum and less so in the colon, whereas *SATB2* is more abundant in the colon than in the ileum. This is consistent with the observation that *SATB2* is significant for UC but not for CD (Table 2). *PPP3CA* expression is not evident from bulk RNA-Seq data of intestinal tissue. **b** *NR5A2* is expressed in ileal progenitor, stem cells, and transient amplifying (TA) cells, less strongly in colon and rectum, and not in the brain, consistent with bulk RNA-seq results. *SATB2* is strongly expressed in all cell types of the colon and rectum. *PPP3CA*, despite weak expression in the ileum or colon, is enriched in enteroendocrine and Paneth-like cells of the colon and rectum, supporting a role in immunomodulation. In the brain, *PPP3CA* is enriched in neurons. Overall, these expression data indicate that separate functions are likely in the gut and brain-based on specific expression patterns. *NR5A2* expression was not found in the brain, but literature results make brain-specific function highly likely (Supplementary Note 3). Raw data of this figure can be found in Supplementary Data 17. (exPFC glutamatergic neurons from the PFC, exCA1/3 pyramidal neurons from the Hip CA region, GABA GABAergic interneurons, exDG granule neurons from the Hip dentate gyrus region, ASC astrocytes, NSC neuronal stem cells, MG microglia, ODC oligodendrocytes, OPC oligodendrocyte precursor cells, NSC neuronal stem cells, SMC smooth muscle cells, END endothelial cells).

is likely that genetic variation mediated by gene expression in the Wnt signaling pathway and NFAT activation by calcineurin is associated with both SCZ and IBD. One limitation is that we could only use candidate genes from six TWAS loci for pathway analysis, so future, more powerful TWAS are expected to provide a more detailed picture here. Another limiting factor of our study is that the heritability of our three (conservatively) selected gene-disease associations accounts for only a small fraction of the estimated $h_{med}^2$ and may present only the tip of the iceberg. We avoided further multiple testing correction for all seven traits, as this would be counterproductive for TWAS screening after Bonferroni correction for the number of all gene-disease associations (although not statistically independent) and conditional analysis for nearby genes at suggestive loci. In addition, the proportion of shared $h_{med}^2$ between diseases is difficult to quantify and requires the development of bivariate (for pairs of diseases) methods to estimate the shared $h_{med}^2$.

A very interesting observation is the gene-disease association with *PPP3CA*. *PPP3CA* encodes a subunit of calcineurin (the catalytic subunit Calcineurin A). Calcineurin is a $Ca^{2+/}$calmodulin-regulated serine/threonine protein phosphatase[49], which activates the transcription factor NFAT (nuclear factor of activated T-cells) and thus plays a key role in the regulation of the immune response (Fig. 5). After multiple-gene-conditioned fine-mapping analyses at loci with multiple-gene-disease association signals, in total, 18 out of 55 genes of the NFAT signaling pathway showed a gene-disease association with either IBD or SCZ, or both (Fig. 5; Methods). Calcineurin inhibitors (e.g., cyclosporine A and tacrolimus) have long been used as immunosuppressants after solid organ transplantation and are used as first-line treatment in PSC patients who have undergone liver transplantation[50]. Calcineurin inhibitors are also being studied for their efficacy in a number of autoimmune diseases[51]. Recent clinical trials with tacrolimus have demonstrated the efficacy and safety of tacrolimus in treatment-resistant UC[52] and ulcerative proctitis (UC confined to the rectum)[53,54]. However, knockout of calcineurin in the forebrain of mice was found to be associated with symptoms of schizophrenia-like psychosis[55], and recent experiments in rats have shown that calcineurin inhibitor therapy may be a risk factor for the development of neurobehavioral changes[56]. In humans, treatment with calcineurin inhibitors sometimes showed an increase of neuropsychiatric side effects[57,58] and tacrolimus has been reported to cause a relapse of schizophrenia[59]. Calcineurin-inhibiting immunosuppression used after solid organ transplantation have also been associated with multiple neuropsychiatric side effects[60]. Consistent with these

results from clinical and functional studies, we found that decreased predicted expression of *PPP3CA* is associated with increased risk for SCZ (Table 2).

Our analysis of bulk RNA-Seq and scRNA-Seq data showed that *PPP3CA* expression is mainly restricted to enteroendocrine and Paneth-like cells of the ileum, colon, and rectum. The gut microbiota produces various metabolites (including short-chain fatty acids, secondary bile acids, and lipopolysaccharides) that modulate enteroendocrine cells and produce hormonal signals reflecting food intake, microbial composition, and epithelial integrity[61]. Paneth cells are secretory epithelial cells of the intestine that contain antibacterial proteins and alpha-defensins that are released into the intestinal lumen in response to a series of stimuli[62], indicating a possible link to the GBA. *PPP3CA* is also expressed in the brain, there is evidence for a general implication in SCZ, and studies show more calcineurin immunoreactive neurons in the caudate nucleus of SCZ patients (Supplementary Note 3), which, together with the putamen where we found the TWAS gene-disease association, forms the striatum. Taken together with the literature (Supplementary Note 3), *PPP3CA* has probably at least two distinct functions relevant to SCZ or IBD, one in neuronal signal transduction in the striatum and the other in signal transduction for immune modulation in Paneth and enteroendocrine cells. *NR5A2* (encoding human LRH-1) regulates intestinal glucocorticoid synthesis and is known to protect epithelial integrity and attenuate inflammatory damage in murine and human intestinal organoids, including those derived from IBD patients (Supplementary Note 3). Although *NR5A2* appears to be nearly absent from the brain and blood based on the present data, there is evidence of its function in brain tissues (Supplementary Note 3), so we suspect that *NR5A2* expression is restricted to less-studied regions of the brain and that it has separate functions in the gut and brain. *SATB2* is a promising candidate for a GBA susceptibility gene because it is predominantly expressed in the colon sigmoideum and prefrontal cortex, consistent with our TWAS results in which *SATB2* expression is associated with SCZ and UC but not with CD. *SATB2* is also a prognostic marker in UC-associated colorectal cancer (Supplementary Note 3). In summary, gene expression analyses for *NR5A2*, *SATB2*, and *PPP3CA* using intestinal and brain tissue data from different developmental stages and from scRNA-seq studies support the findings from previous human, mouse, and organoid studies and suggest a pleiotropic role for these genes as part of the GBA.

## Methods

**Consortium GWAS summary statistics.** Crohn's disease (CD) and ulcerative colitis (UC) case and control cohorts (Supplementary Data 1) from 15 countries

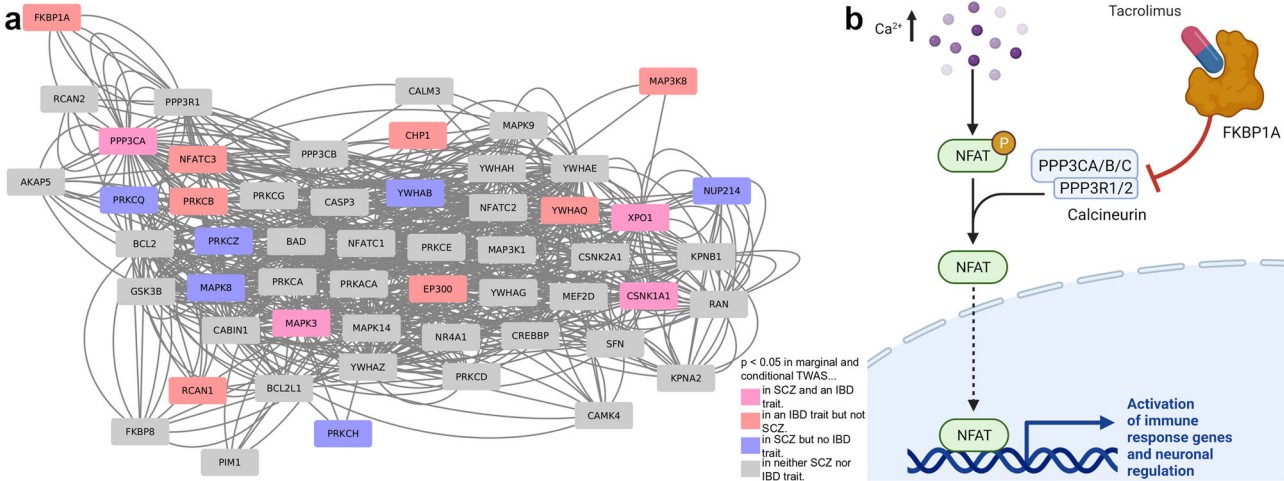

**Fig. 5 Calcineurin-dependent NFAT signaling as shared signaling pathway for IBD and SCZ. a** 18 gene-disease associations with suggestive significance ($P_{GBJ\ conditional} < 0.05$; i.e., after multiple-gene-conditioned fine-mapping analysis) are enriched in the "calcineurin-dependent NFAT signaling in leukocytes" pathway (comprising 55 genes; Methods), showing genetically altered gene expression for SCZ, CD, or UC compared to healthy controls. *PPP3CA* encodes the catalytic subunit calcineurin A; *FKBP1A* (also known as *FKBP12*) encodes a cis-trans prolyl isomerase that binds the immunosuppressant FK506 (tacrolimus). **b** The calcineurin inhibitor tacrolimus, used against various inflammatory diseases and as an immunosuppressant in organ-transplanted PSC-IBD patients, prevents NFAT signaling by binding to FKBP1A and causes inhibition of calcineurin. The reduced genetic expression of *PPP3CA* that we identified in intestinal and brain tissues causes an increased risk of SCZ and CD (Table 2). Raw data of this figure can be found in Supplementary Data 18.

across Europe, North America, and Australia have previously been described, quality controlled (QCed), and genotype imputed[13,32,63]. Primary sclerosing cholangitis (PSC) case and control cohorts from 14 countries in Europe and North America and have previously been described, QCed and imputed[13,64]. The number of overlapping (duplicate) GWAS samples for CD, UC, and PSC GWAS data sets were determined by identity-by-descent (IBD) analysis[13]. Because of a partial sample overlap in cases and controls between Immunochip and GWAS data sets for CD, UC, and PSC (Supplementary Data 1), the smaller sample-sized GWAS data sets for CD, UC, and PSC (data sets #2, #4, #6 in Supplementary Data 1) were used only for validation of TWAS association results from the TWAS discovery phase (see text below). Schizophrenia (SCZ), major depressive disorder (MDD), bipolar disorder (BD), and attention-deficit/hyperactivity disorder (ADHD) imputed GWAS summary statistics were used as described elsewhere[65–68] and were downloaded from https://www.med.unc.edu/pgc/download-results/.

Written, informed consent was obtained from all study participants, and the institutional ethical review committees of the participating centers approved all protocols.

**Selection of tissues for TWAS of the GBA**. Recently, a heritability enrichment analysis of specifically expressed genes across 205 tissues and cell types in combined with GWAS summary statistics of CD, UC, SCZ, MDD, BD, and ADHD and subsequent validation of enrichments results with chromatin data yielded statistically significant enrichment results for tissues of the central nervous system (CNS) for psychiatric diseases and for tissues from blood/immune cell types and the gastrointestinal tract for inflammatory bowel diseases[19]. Based on these results, we restricted our TWAS analyses using genome-wide and transcriptome-wide reference data of 4043 samples from brain, immune cell and the gastrointestinal tissues (Supplementary Data 2) available from consortia projects GTEx[20] (version V7, dbGaP accession code phs000424.v7.p2), STARNET[21] (dbGaP accession code phs001203.v1.p1), and BLUEPRINT[22] ftp://ftp.ebi.ac.uk/pub/databases/blueprint/). As described by Hu et al.[16], biallelic SNPs with minor allele frequency (MAF) ≥0.01 were retained and normalized gene expression information was adjusted to correct for potentially confounding effects of sex, sequencing platform, and the three main principal components from principal component analysis (PCA) of the genome-wide genotype data, and with an estimation of expression residuals (PEER) factors[69].

**UTMOST expression imputation models**. To estimate the genetically regulated expression of each gene in the genome across 23 tissues of the GBA, we applied multivariate-response penalized regression models as implemented in UTMOST (https://github.com/Joker-Jerome/UTMOST) to predict cross-tissue gene expression from reference data sets. Briefly, a standard least-squares loss function was minimized, with an L1 Lasso penalty adaptively set based on sample sizes to select predictive SNP variables in a range of ±1 Mb around each gene and force shrinkage in effect size estimation (within-tissue effects), and further with a group-Lasso penalty set across multiple tissues on the effect sizes of each SNP to select eQTLs shared between tissues (cross-tissue effects)[16]. Such cross-tissue models for expression imputation favor eQTLs that are shared across tissues while preserving

tissue-specific effects, and UTMOST's approximation procedure proposed for coordinate descent iteration handles incomplete data in the reference expression matrix.

**Association testing and GBJ test**. UTMOST was further used to test for gene-based association at the level of gene expression regulation, taking into account cross-tissue regulatory effects. A univariate regression model was applied to test the association between predicted gene expression (i.e., the genetically regulated component of gene expression) and disease status for each tissue separately (although previously trained over several tissues at the same time). To summarize the tissue-specific results and test whether a gene is significant for at least one tissue, a generalized Berk–Jones (GBJ) test[23] was used to combine associations across individual tissues into a unified test of association. The GBJ is based on the absolute values of gene trait Z-score, allowing for directions of eQTL effects in the different tissues, and uses tissue covariance to correctly weight results across tissues and correct for multiple tissue testing. We used the Bonferroni correction on the number of genes tested to determine transcriptome-wide significance ($P < 0.05/15,587 = 3.20 \times 10^{-6}$). For the CD, UC, and PSC phenotypes, it had to hold additionally that transcriptome-wide significant genes replicate with $P < 0.05$ in the smaller GWAS data sets #2, #4, #6 in Supplementary Data 1.

**Comparison with other pre-trained single-tissue expression imputation models**. The elastic net approach from S-PrediXcan[24] was used for expression imputation of 19 GTEx tissues (Supplementary Data 2) followed by gene-disease association testing using logistic regression for our ten study data sets (Supplementary Data 1). The precomputed reference based on GTEx v7 data was available at http://predictdb.org/. In addition, FUSION's[25] Bayesian linear mixed model was used for expression imputation and gene-disease association testing with precomputed weights derived from the same 19 GTEx tissues v7 (http://gusevlab.org/projects/fusion/). The number of Single-tissue_marginal significant genes ($P < 3.20 \times 10^{-6}$) from UTMOST, FUSION, and S-PrediXcan for the 19 GTEx tissues and the 10 GWAS studies included in this study were compared with the number of reference samples used for gene expression imputation training. The number of reference samples used in expression imputation training was taken from the supplementary information or the databases of the respective publications; it is possible that the number of GTEx v7 reference samples differs for the different tools due to different pre-filtering analyses. For all tools, only the marginal, uncorrected, P values were considered for benchmark purposes. Spearman correlation and a simple linear regression model were computed separately for each of the ten studies. A paired Mann–Whitney U-test was applied to test for differences in slopes (β) between the three tools.

**Cross-tissue multiple-locus-genes conditional analysis and GBJ test**. We used the UTMOST multiple regression approach to perform multiple-locus-genes conditional cross-tissue analysis for each transcriptome-wide significant gene (from unconditioned analysis) in each of the ten studies separately to prioritize gene-disease associations at the same locus within the TWAS locus boundaries of the

±1 Mb range. The conditional analysis accounts for one of the main problems of the TWAS gene-based association test, namely the potential co-regulation of multiple genes by the same eQTLs or the presence of independent eQTLs across genes that are in linkage disequilibrium (LD)[15]. Again, the association statistics of the conditional analysis were combined across tissues using the generalized Berk–Jones score (GBJ) test described above. Again, we used the Bonferroni correction on the number of genes tested to determine transcriptome-wide significance ($P < 0.05/15{,}587 = 3.20 \times 10^{-6}$).

**Overlap of gene-disease associations with the boundaries of known susceptibility loci from genome-wide association studies (GWAS) for CD, UC, PSC, SCZ, MDD, BD, ADHD.** For genomic position comparison, locus boundaries of established GWAS loci were adopted from the last fine-mapping GWAS studies for SCZ[31] and CD/UC[32,33]. A gene was labeled "Overlap with GWAS locus" in Table 2 if the gene had an overlap with an established GWAS locus. Gene coordinates (GRCh37, Jan 2020) were obtained using the Ensembl database and BioMart[70].

**Comorbidity analysis in the Danish National Patient Registry (DNPR).** To determine significant co-occurrences (disease pairs) for the seven diseases under study, we screened an independent dataset covering ICD-10 diagnosis codes from 7,191,385 people of the entire Danish population in the period from 1994 to 2018[71]. The DNPR includes primary and secondary diagnoses coded according to the International Statistical Classification of Diseases 10th Revision (ICD-10) from all Danish hospitals. We identified patients diagnosed with one of the seven diseases, for example, CD. Subsequently, we identified a random control group of 20 controls, matched on same-sex and year of birth for each patient diagnosed with CD. Since some of the psychiatric disorders we studied have an average prevalence of more than 1% in the general population, which is especially true for depression and ADHD, we needed a sufficiently high number of controls per case to avoid a significant number of cases occurring by chance in the control group. However, the gain in statistical power, in general, decreases rapidly beyond 4 to 20 controls per case[72]. Therefore, we have selected 20 controls per case so that the calculation does not take too long and because additional controls would not result in a significant gain of statistical power.

The incidence of the six other diseases ($X$) was calculated for the patients with CD and the randomly matched control group and the strength of the association between the diseases was assessed by relative risk (RR) using equation 1. Here, $A_x$ is the number of CD patients also diagnosed with disease $X$, while $A_t$ is the total number of CD patients. Likewise, $C_x$ is the number of controls diagnosed with disease $X$ and $C_t$ is the total number of control patients.

$$\text{Relative risk} = \frac{A_x/A_t}{C_x/C_t} \qquad (1)$$

$P$ values for all disease pairs were calculated using a two-sided Fisher exact test and corrected for multiple testing using the false discovery rate (FDR). This analysis is repeated for all seven diseases under study and $A_t$, $A_x$, RR as well as FDR adjusted $P$ values were reported in Supplementary Data 8.

**Trait correlation at the level of GWAS summary statistics (LDSC).** We applied LDSC[73] genome-wide correlation as described in Tylee et al.[11]. Shortly, we excluded the extended MHC region and filtered for HapMap3 SNPs with an INFO score of at least 0.9 and a MAF of at least 0.05. The summary stats were subsequently processed by the munge_sumstats.py program of the LDSC software package (https://github.com/bulik/ldsc), and heritabilities and genetic correlation were calculated on the liability scale using population prevalences (Supplementary Data 9).

**Trait correlation at the level of predicted gene expressions.** Genes are often co-regulated, i.e. they either share the same eQTLs or eQTLs exist in strong LD. For this reason, we used a subset of 1825 co-regulated and eQTL-independent genes recently provided for Mendelian randomization studies[28], and further excluded genes from the extended MHC region (chr6:25-34 Mb) to avoid inflated correlation measures. Spearman's correlation of all traits with each other was calculated separately for each of the 23 tissues based on the single-tissue marginal $Z$-scores (Supplementary Data 10) and additionally based on the GBJ test marginal test-statistics (Supplementary Data 10 and Fig. 2).

**Mendelian randomization (MR) analysis at the level of predicted gene expressions.** We performed multi-gene MR analysis at the level of predicted gene expressions using GSMR[30] (Generalized Summary-data-based Mendelian Randomization) to test for a causal association between one trait (exposure) on another (outcome) using summary-level TWAS gene association statistics. We used only standardized transcriptome-wide significant marginal UTMOST gene-disease associations as instruments ($p_{marginal\&conditional} < 3.2e\text{-}6$) that were likely independent of neighboring genes by enforcing a minimum distance of 1 Mb between the instrumental genes (see section *Cross-tissue multiple-locus-genes conditional analysis*). MR analysis was performed on our tissue-specific TWAS results. At least

10 significant, independent gene-disease associations were necessary as instruments to obtain reliable regression results[30]. Because the genetically regulated expression is (by definition) heritable and inferred from SNP-disease association effects, the MR approach from GSMR was performed with (statistically independent) TWAS genes. This satisfies the requirements of MR analysis according to Bowden et al.[74]. We further used the HEIDI ((heterogeneity in dependent instrument)-outlier test)[30] to detect and eliminate pleiotropic gene-disease associations (genetic instruments) that have apparent pleiotropic effects (horizontal pleiotropy, $P_{HEIDI} < 0.01$) on both traits (exposure and outcome) under investigation.

**Gene overlap analysis.** To find potential genes mediating the gut-brain axis in the same or different tissues, we considered as candidates all genes that were transcriptome-wide significant for both one inflammatory and one psychiatric trait (i) in the same tissue or (ii) in the GBJ test, both marginally ($P_{marginal} < 3.20 \times 10^{-6}$) and conditionally ($P_{conditional} < 3.20 \times 10^{-6}$) significant. Option (ii) is more liberal and thus includes genes, which are significant in one tissue in a psychiatric trait but significant in an inflammatory trait in another tissue.

**Gene expression heritability estimation.** Because the heritability of gene expression of each gene was not given by UTMOST expression reference and training files, we extracted heritability estimates of all available tissues from our FUSION results (see section Comparison with other pre-trained single-tissue expression imputation models) by determining the lowest and the highest heritability value per gene, respectively, to give a kind of confidence range of the expected heritability. Briefly, for heritability estimation, the *cis* locus was defined as ±500 kb of gene boundaries. Samples were restricted to Europeans by principal components analysis (PCA). FUSION's heritability models were adjusted for 20 gene expression principal components from PCA, two genetic ancestry PCs, and local structural variation estimated from SNP array data of reference data sets[25]. Heritability estimation was performed using GCTA/GREML[75].

**GWAS/TWAS conditional analysis.** If a significant TWAS disease gene is present for a given locus, we expect that the eQTLs of this gene explain a portion of the original GWAS signal seen in the input GWAS summary statistics, for example, in the presence of statistically causal protein-coding variants. To estimate the proportion of the GWAS signal explained by the eQTLs of a significant TWAS candidate gene, GWAS summary statistics were conditioned with GCTA COJO[76], version 1.92.2) using a maximal set of noncolinear eQTL variants of a TWAS gene (of ±1 Mb flanking region from the gene) for each candidate gene of interest. The aim of this analysis was to qualitatively test whether the eQTLs of a transcriptome-wide significant gene were a possible statistical explanation for the GWAS association signal. GCTA COJO uses the allele frequency, beta, standard error of the beta, the association $P$ value, and the population size of the GWAS summary statistics as input. All eQTLs with a nonzero effect of a significant TWAS candidate gene were given to COJO using parameter --select, whose algorithm is designed to find noncolinear associated SNPs (MAF > 0.01) to subsequently condition on. The $P$ value threshold for including disease-associated noncolinear eQTL sets was chosen 0.05; this relatively weak association significance threshold was chosen to additionally identify eQTLs from non-genome-wide significant regions. The resulting set of noncolinear eQTL SNPs was used to condition the GWAS summary data using the GCTA COJO parameter --cond via conditional linear regression analysis[34]. The resulting TWAS-conditioned GWAS summary statistic represents the minimal portion of the GWAS signal that is not covered by the eQTLs of the selected gene and thus could not have been included in the TWAS association. This may suggest that the TWAS gene, although significantly associated, may not be sufficient to fully explain the GWAS association at this locus.

In summary, our complete UTMOST- and GCTA COJO-based analysis can answer three questions on the association of genetically regulated expression of a gene in a given tissue; (i) whether the eQTLs weighted from the UTMOST-trained models provide an explanation for a GWAS signal with ±1 Mb around a TWAS candidate gene (marginal UTMOST test), (ii) which TWAS gene is most likely to represent a GWAS signal given multiple associated TWAS genes (conditional UTMOST test), and (iii) whether there are SNP-disease associations in the GWAS data that cannot be explained by eQTLs in LD (COJO test), such that there may be a different eQTL gene, causal (protein-coding) variants, or epigenetic modifications at the locus that result in a GWAS association signal.

**Bulk RNA-Seq data of different developmental stages and scRNA-seq expression data lookup for tissues of the GBA.** Median TPM- (transcripts per million) normalized bulk tissue gene expression data from 49 tissues of 838 postmortem donors were retrieved from the data portal of GTEx project[77] (https://www.gtexportal.org/home/datasets) including (i) a dataset of a total of 297 mRNA libraries from 23 developmental stages from human fetal (elective abortion) to unrelated deceased postnatal, juvenile and adult tissue samples[38] (E-MTAB-6814, https://www.ebi.ac.uk/gxa/experiments/E-MTAB-6814/Downloads), (ii) a dataset of 50 mRNA libraries from the expression of the human fetal (elective abortion) and juvenile gut biopsies, two different tissues each, from intestinal epithelial organoids and purified intestinal epithelial cells[39] (E-MTAB-5015, https://www.ebi.ac.uk/gxa/experiments/E-MTAB-5015/Downloads) and (iii) the

BLUEPRINT blood cell reference data from 77 mRNA libraries of 27 cell types from infant to adult donors (E-MTAB-3827, https://www.ebi.ac.uk/gxa/experiments/E-MTAB-3827/Downloads). The obtained gene expression values were gene-wise centered and Z-score normalized for visualization in R.

Single-cell RNA-seq (scRNA-seq) data of healthy human small intestine, colon, and rectum biopsies, from two donors each, with a total of 14,537 cells were retrieved from GSE125970[41] and single-cell libraries from five male postmortem donors (three prefrontal cortex samples, four hippocampus) from the GTEx biobank (Habib et al.; 28846088) were retrieved via https://www.covid19cellatlas.org. Data were processed and log-normalized as described by Sungnak et al.[78] and visualized using the scanpy v1.6.0 package[79] (https://scanpy.readthedocs.io/en/stable/api/scanpy.pl.dotplot.html).

**NFAT pathway visualization**. The list of 55 genes of the term "calcineurin-dependent NFAT signaling role in lymphocytes" from the database BioPlanet 2019[36] was downloaded from https://maayanlab.cloud/Enrichr/#stats[80] (May 25, 2021) and used as input for GeneMania[81] with no genes added and all available physical interaction edges, pathway-based edges and colocalization edges plotted. The resulting network was exported and imported into Cytoscape 3.8.2[82] to adjust the visual style. We chose both marginal and conditional nominal significance ($P_{GBJ\ conditional} < 0.05$) as the threshold for highlighting genes.

**Reporting Summary**. Further information on research design is available in the Nature Research Reporting Summary linked to this article.

## Data availability
The source data used for this study can be downloaded from the following repositories: The GWAS data can be found as PGC GWAS summary statistics (https://www.med.unc.edu/pgc/download-results/) and IIBDGC GWAS summary statistics (ftp://ftp.sanger.ac.uk/pub/consortia/ibdgenetics/iibdgc-trans-ancestry-filtered-summary-stats.tgz). Gene expression data can be found at BLUEPRINT (ftp://ftp.ebi.ac.uk/pub/databases/blueprint/), GTEx, (https://www.gtexportal.org/home/datasets), gene expression atlas, accession numbers E-MTAB-6814 (https://www.ebi.ac.uk/gxa/experiments/E-MTAB-6814/Downloads), E-MTAB-5015, (https://www.ebi.ac.uk/gxa/experiments/E-MTAB-5015/Downloads), E-MTAB-3827, (https://www.ebi.ac.uk/gxa/experiments/E-MTAB-3827/Downloads), and at the COVID-19 cell atlas https://www.covid19cellatlas.org. Gene ontology enrichment reference data of BioPlanet 2019 were downloaded from the Enrichr website https://maayanlab.cloud/Enrichr/#stats. The data availability of these data is not subject to any restrictions.

## Code availability
Codes used in this paper are available from websites of the ScanPy package (https://scanpy.readthedocs.io/en/stable/api/scanpy.pl.dotplot.html), the UTMOST software (https://github.com/Joker-Jerome/UTMOST), the MetaXcan software, (https://github.com/hakyimlab/MetaXcan, http://predictdb.org/), the FUSION software (http://gusevlab.org/projects/fusion/), and the LDSC software (https://github.com/bulik/ldsc).

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

## Acknowledgements

This work was supported by the German Federal Ministry of Education and Research (BMBF) within the framework of the e:Med Vernetzungsfonds funding concept (GB-XMAP grants 01ZX1709A, 01ZX1709B, and 01ZX1709C). The study received infrastructure support from the Deutsche Forschungsgemeinschaft (DFG, German Research Foundation) Cluster of Excellence 2167 "Precision Medicine in Chronic Inflammation (PMI)" (EXC 2167-390884018). Access to the DNPR was approved by the Danish Health Data Authority (FSEID-00003092 and FSEID-00004491) and supported by Novo Nordisk Foundation (grants NNF14CC0001 and NNF17OC0027594). We thank the PGC for making the GWAS summary statistics publicly available. We thank the IIBDGC and IPSCSG for permission to use the GWAS/Immunochip summary statistics; a complete list of members and affiliations of the IIBDGC and the IPSCSG can be found in the Supplementary Information. Figure 5b was generated using BioRender.com and a template from Akiko Iwasaki, Ph.D. and Ruslan Medzhitov, Ph.D.

## Author contributions

DE was responsible for the concept and the design of the study. TF, AF, MN, FD, THK, SS, CS, PH, and PK assisted with recruitment or phenotyping of patients in the inflammatory and psychiatric disease clinical cohorts from which the GWAS summary statistics were derived or provided GWAS summary statistics. FU-W and DE implemented statistical models and performed computation analyses. CM, OB, EMW, IFJ, S Be, OW, KB, and S Br. helped with bioinformatic analyses. SJ and FU-W performed RNA-sequencing data analysis. TAS designed Fig. 1. DE and FU-W curated and interpreted results and wrote the manuscript. All authors reviewed, edited, and approved the final manuscript.

## Funding

## Competing interests

The authors declare no competing interests.
