## [Peer Review File · Communications Biology]

Reviewers' comments:

Reviewer #1 (Remarks to the Author):

This is a well-written paper with many interesting results. The data analysis methods are very solid. Specifically, it performed cross-tissue and multiple-gene conditioned transcriptome-wide association studies (TWAS) for 23 tissues of the gut-brain-axis (GBA) using GWAS data sets (total 246,772 46 patients and 638,454 controls) for Crohn's disease (CD), ulcerative colitis (UC), primary 47 sclerosing cholangitis (PSC), schizophrenia (SCZ), bipolar disorder (BD), major depressive 48 disorder (MDD) and attention-deficit/hyperactivity disorder (ADHD). The authors are able to identify NR5A2, 49 SATB2, and PPP3CA as shared susceptibility genes with transcriptome-wide significance both for CD/UC and SCZ, largely explaining fine-mapped association signals at nearby GWAS susceptibility loci. I only have a small comment regarding the following analysis.

1. The authors identified patients diagnosed with one of the seven disease and then a control group of 20 controls for each case, matched by sex and year of birth. Could the authors elaborate more on such setting? Furthermore, it is unclear whether the selection of 20 controls could influence the strength of the association between the diseases, relative risk (RR), and P-values for a two-sided Fisher exact test.

Reviewer #2 (Remarks to the Author):

Review of "Cross-tissue transcriptome-wide association studies of 885,176 individuals and seven diseases of the gut-brain axis identify susceptibility genes shared between schizophrenia and inflammatory bowel disease"

The authors conduct TWAS analyses to exploring the sharing of genetic mechanisms across immune-related and psychiatric traits. They conduct fine-mapping analyses to identify the causal genes most likely to underlie the TWAS associations. Analyses have been conducted using a wide range of RNA-seq reference datasets from various resources. Single and multi-tissue analyses have been conducted. Overall, I consider this to be a high quality study, although the discussion of findings should consider the polygenic nature of the traits (especially in the context of psychiatric disorders). I believe that some of the findings are overstated, as the proportion of genetic/phenotypic variance accounted for by the highlighted genes is likely to be small. I have some specific comments/questions that will hopefully help the readers to further strengthen the manuscript.

Major comments:

1) From introduction, it is not clear whether genetic overlap (e.g., LDSC, LAVA) analyses have been conducted. If so, results should be described in the introduction. If not, TWAS analysis should be complemented with genetic correlation analyses. The authors later reference some studies that explored genetic overlap. It would be helpful for the reader if results are briefly presented in the introduction

2) The authors should explicitly note that there was no multiple testing correction across traits and refer to this as a potential limitation. I don't disagree with the author's decision per se, but it would need to be clear to the reader. The discussion does not currently have a limitations section; I think it should be added as there are a number of important limitations (as for every study).

3) Is spearman correlation the most appropriate measure for calculating the TWAS correlations? (line 176-180). I understand the number of genes is limited based on a previously determined set, but think it may be good to compare with rho correlation analysis that has been specifically developed for TWAS

<https://www.ncbi.nlm.nih.gov/pmc/articles/PMC5339290/>

<https://github.com/bogdanlab/RHOGE>

4) The authors highlight 3 genes that overlap between immune and psychiatric traits. (lines 222-226). It is not clear to me what the strength of the statistical evidence is supporting these genes. The statistical evidence is obtained from the univariable (trait) analyses, but how many genes would be expected to show significant associations in two or more traits based on chance? The number of comparisons is large. In fact, the total number of genes (3) that show significant association across traits and across brain and intestinal tissues seems low to me. To what extent

does this support the hypothesis of genetic overlap? (at a TWAS level). The authors refer to the gene expression heritability estimates, but although these are relevant, they form only a basis for significant TWAS association. These numbers are again reported in discussion, but do not provide insight into how much of the phenotypic/genetic variation in the traits is explained by these genes.

5) Throughout the paper, there are sentences where the information is so dense that they are difficult to read. I will give one example, but the entire manuscript should be checked. lines 229-231. I think that the authors imply that NR5A2 and SATB2 are associated with SCZ, CD and/or UC are in the same direction, while PPP3CA is associated in opposite directions. If this is true, what are the implications? (Tables and Figures actually suggest my interpretation was incorrect, still showing that the original sentence was difficult to read).

6) Lines 303-306: A gene can only have a significant TWAS association if there is expression (and variation in expression). Therefore, I am not sure this info is very relevant. It is using the same data as the TWAS, so findings are not surprising at all.

7) Lines 411-414: I am not very convinced or excited by this entire paragraph, as the authors overstate findings by ignoring the polygenicity of the traits. These sentences are especially problematic as the clinical implications described here are not supported sufficiently by their results. How much of the genetic/phenotypic variation in the traits is explained by the highlighted genes?

8) The authors seem to have used outdated SCZ Summary stats although they do cite the latest one (Pardinas). The reason for this is unclear.

9) The analyses for ADHD, MDD, and BD seem to be relatively underpowered. This should be noted as a limitation. The authors could consider to use Howard et al. for the MDD GWAS (although these are not all clinical cases).

10) From the main text, it seemed that the 3 key genes were associated across brain and intestinal tissues. From Table 2 it becomes clear that the associations for SCZ only exist in brain. This seems to contradict the main conclusion of the authors that these pleiotropic genes influence immune and psychiatric traits through their role in brain and intestinal tissues?

11) Overall, the quality of figures and tables is good. However, Figure 3 is difficult to understand. Even after reading the legend several times, I still do not shown what information is shown below the manhattan plots (green circles and green bars). It should be explained in the figure itself. The other information was clear in the figure labels, but not so much in the legend. Also, in figure 3, the * (TWAS gene eQTLs are difficult/impossible to detect).

12) Figure 5: how much of the genetic/phenotypic variance is explained by this pathway? (or the genes in this pathway). Could the authors conduct a more formal enrichment analysis?

Minor comments:

- Line 111: "Gene expression imputation was conducted using GWAS summary statistics". I'm not sure if I would refer to this as imputation as no raw genotype data (individual-level) were available
- Line 114: combining N controls across traits is not very informative.
- Line 182: reference to CD4+ T-cells is confusing, was this not a genome-wide analysis?
- Line 187-190: Can the authors statistically validate this statement?
- Line 212: Why was the number of genes 171 for the MR analysis and 288 for this analysis?
- Line 247: typo, GTCA should read GCTA
- Line 277: How can a gene be implicated by a non-genetic study?
- Line 572-573: I think this would also need to explained in the GJB test section
- The font size and quality of Figure 2B should be improved

Point-by-point reply for Submission COMMSBIO-21-1764-T, “Cross-tissue transcriptome-wide association studies of 885,176 individuals and seven diseases of the gut-brain axis identify susceptibility genes shared between schizophrenia and inflammatory bowel disease”

Reviewer Comments:

Reviewer 1 (Remarks to the Author):

This is a well-written paper with many interesting results. The data analysis methods are very solid. Specifically, it performed cross-tissue and multiple-gene conditioned transcriptome-wide association studies (TWAS) for 23 tissues of the gut-brain-axis (GBA) using GWAS data sets (total 246,772 46 patients and 638,454 controls) for Crohn’s disease (CD), ulcerative colitis (UC), primary 47 sclerosing cholangitis (PSC), schizophrenia (SCZ), bipolar disorder (BD), major depressive 48 disorder (MDD) and attention-deficit/hyperactivity disorder (ADHD). The authors are able to identify NR5A2, 49 SATB2, and PPP3CA as shared susceptibility genes with transcriptome-wide significance both for CD/UC and SCZ, largely explaining fine-mapped association signals at nearby GWAS susceptibility loci.

We thank the Reviewer for the supportive feedback and interest in our study.

I only have a small comment regarding the following analysis.

1. The authors identified patients diagnosed with one of the seven disease and then a control group of 20 controls for each case, matched by sex and year of birth. Could the authors elaborate more on such setting? Furthermore, it is unclear whether the selection of 20 controls could influence the strength of the association between the diseases, relative risk (RR), and P-values for a two-sided Fisher exact test.

We now described our approach in more detail in the Methods section “Comorbidity analysis in the Danish National Patient Registry (DNPR)”. First, we identified patients in the Danish National Patient Registry (DNPR) who were diagnosed with one of the seven diseases, for example Crohn’s disease (CD), in Danish hospitals. Then, for each patient diagnosed with CD, a random control group of 20 individuals of the same sex and birth was further identified. The incidence of the six other diseases (X) for the period 1994 to 2018 was calculated for the CD patients and the randomly matched control group, and the strength of the association between pairs of diseases was assessed by relative risk (RR) using equation 1 (added to the Methods section). P-values for all disease pairs were calculated using a two-sided Fisher exact test and corrected for multiple testing using the false discovery rate (FDR). This analysis was repeated for all seven diseases (Supplementary Table 8).

Since some of the psychiatric disorders we studied have an average prevalence of more than 1% in the general population, which is especially true for depression and ADHD, we needed a sufficiently high number of controls per case to avoid a significant number of cases occurring by chance in the control group. However, the gain in statistical power in general decreases rapidly beyond 4 to 20 controls per case (Pang et al., 1999, PMC1757658). Therefore, we have selected 20 controls per case so that the calculation does not take too long and because additional controls would not result in significant gain of statistical power.

Reviewer #2 (Remarks to the Author):

Review of “Cross-tissue transcriptome-wide association studies of 885,176 individuals and seven diseases of the gut-brain axis identify susceptibility genes shared between schizophrenia and inflammatory bowel disease”

The authors conduct TWAS analyses to exploring the sharing of genetic mechanisms across immune-related and psychiatric traits. They conduct fine-mapping analyses to identify the causal genes most likely to underlie the TWAS associations. Analyses have been conducted using a wide range of RNA-seq reference datasets from various resources. Single and multi-tissue analyses have been conducted. Overall, I consider this to be a high quality study, although the discussion of findings should consider the polygenic nature of the traits (especially in the context of psychiatric disorders). I believe that some of the findings are overstated, as the proportion of genetic/phenotypic variance accounted for by the highlighted genes is likely to be small. I have some specific comments/questions that will hopefully help the readers to further strengthen the manuscript.

We are very grateful to Reviewer #2 for the constructive suggestions to improve the presentation of our results and the readability of the manuscript. We have highlighted all changes in the manuscript via Word tracked changes.

Major comments:

1) From introduction, it is not clear whether genetic overlap (e.g., LDSC, LAVA) analyses have been conducted. If so, results should be described in the introduction. If not, TWAS analysis should be complemented with genetic correlation analyses. The authors later reference some studies that explored genetic overlap. It would be helpful for the reader if results are briefly presented in the introduction

We have now moved the reported results of the LDSC study by Tylee et al. 2018 (i.e. significant genome-wide correlation between inflammatory (CD, UC) and psychiatric traits (SCZ, BD)), from the Results section to the Introduction. We have further recalculated the results from the Tylee et al study ourselves using LDSC, because PSC was not studied in Tylee et al. Our own LDSC results widely replicate the results of Tylee et al and we have now added them as a new additional table (Supplementary Table 9). Slightly different LDSC results compared to Tylee et al. are as follows: The genetic correlation between BD and CD ($p=0.055$) as well as UC ($p=0.31$) did not reach significance in comparison to Tylee et al. One reason may be that Tylee et al. used older and smaller sample-sized BD summary statistics (Hou et al., 2016, PMID 27329760) than us (Stahl et al., 2019, PMID 31043756). Tylee et al. already reported in their study that there were some studies before their studies that also found no significant genetic correlations between BD and CD/UC.

2) The authors should explicitly note that there was no multiple testing correction across traits and refer to this as a potential limitation. I don't disagree with the author's decision per se, but it would need to be clear to the reader. The discussion does not currently have a limitations section; I think it should be added as there are a number of important limitations (as for every study).

We thank the reviewer for pointing out the need to introduce a section with limitations of our study. We have now added a "limitations" paragraph (page 13 and 14) to the Discussion section.

Since we can exclude an overlap of patients and no significant sample overlap is to be expected in the controls (see sample overlap calculation in the Methods section), another Bonferroni correction across all seven traits would be counterproductive for TWAS screening and too stringent in terms of statistical power in our view. We have applied a stringent Bonferroni correction for the number of gene-disease associations for the number of all genes, although we could show in our conditional TWAS analyses that in most cases TWAS lead genes (due to LD) are not statistically independent (due to correlated gene expression with nearby genes within a region of 1MB).

Among other limitations (low statistical TWAS power for BD, lack of bivariate methods to estimate shared expression-mediated heritability h_{med}^2 ; see answers below and new

paragraph in the Discussion section), we included the following sentence in the discussion of limitations of our study: “We avoided further multiple testing correction for all seven traits, as this would be counterproductive for TWAS screening after Bonferroni correction for the number of all gene-disease associations (although not statistically independent) and conditional analysis for nearby genes at suggestive loci.”

3) Is spearman correlation the most appropriate measure for calculating the TWAS correlations? (line 176-180). I understand the number of genes is limited based on a previously determined set, but think it may be good to compare with rhoGE correlation analysis that has been specifically developed for TWAS <https://www.ncbi.nlm.nih.gov/pmc/articles/PMC5339290/> <https://github.com/bogdanlab/RHOGE>

Following the Spearman's rank correlation analysis of Pickrell et al. 2016 (PubmedID 27182965) to test for genetic correlation of (independent) GWAS variants in disease pairs, we performed the Spearman's rank correlation analysis for the TWAS summary statistics (our Figure 2b). The idea is the following: In our TWAS search for gene-disease association that influence pairs of phenotypes, for the null hypothesis we assumed no relationship between the effect sizes (i.e. Z scores) of a gene on two different diseases. However, if two traits are influenced by shared underlying molecular mechanisms, we might expect the effects of a gene on the two phenotypes to be correlated. To calculate TWAS correlations between pairs of diseases only for independent gene-disease associations, similar to what is done in the LDSC approach at the genome-wide level for genetic variants (PubmedID 26414676), we used an expression-correlation-pruned list of 1,825 co-regulation- and eQTL-independent genes previously used in Mendelian randomization studies (PubmedID 31341166; see Methods). The results of this Spearman's rank correlation analysis gave us a lower bound (because “unaffected” (non-significant) genes are also included) of which diseases look more similar at the transcriptome-wide level of predicted genetically-regulated expression.

To compare our correlation results with the RHOGE method of Mancuso et al (PubmedID 28238358), which first estimates approximately independent LD blocks and then uses them for correlation calculation, we now performed an RHOGE analysis on the single tissue level. Unfortunately, the RHOGE software (a) only works with the TWAS output results from the TWAS/FUSION software (“Currently, only FUSION style output is supported.”, see <https://github.com/bogdanlab/RHOGE>) and (b) relies on the estimated local heritability calculated by TWAS/FUSION, which thus makes a direct comparison with our results from cross-tissue analysis with UTMOST (Single-tissue_{marginal} analysis but cross-tissue training with UTMOST) difficult. Nevertheless, we ran RHOGE using our TWAS/FUSION results from our Benchmark (Supplementary Table 4, Supplementary Fig. 3).

The RHOGE correlation results correlate well with our Spearman correlation results. (Pearson's rho = 0.79, $p < 2.2e-16$; see figure below). We think that the remaining difference can be explained by the different inputs (FUSION vs. UTMOST), and by the fact that both methods select input genes differently (list of 1,825 co-regulation- and eQTL-independent genes versus independent LD blocks). Because RHOGE cannot process UTMOST results, we stick to our (conservative) Spearman correlation approach. Another disadvantage of RHOGE in our view is that it cannot be applied to GBJ test results from across tissues. Further we found no significant correlation between a psychiatric and an inflammatory trait in RHOGE with the optional RHOGE P-threshold $p < 0.001$, suggesting that the power for significantly associated genes and thus smaller input is not better compared to Spearman correlation. Because of the non-direct comparability of the TWAS-FUSION results (with somewhat higher expected false positive results, see Supplementary Fig. 3) with those of our cross-tissue UTMOST analysis, we prefer not to publish the RHOGE results.

4) The authors highlight 3 genes that overlap between immune and psychiatric traits. (lines 222-226). It is not clear to me what the strength of the statistical evidence is supporting these genes. The statistical evidence is obtained from the univariable (trait) analyses, but how many genes would be expected to show significant associations in two or more traits based on chance?

We have now performed a gene enrichment analysis (Fisher test) to calculate the probability of disease-gene associations shared between pairs of diseases. From the perspective of an enrichment analysis, the chance of detecting one shared disease-gene association for each finding in the respective tissue did not represent a statistically significant result here (enrichment P values from Fisher's test are $P_{\text{Fisher}}=0.06$ for *SATB2*, $P_{\text{Fisher}}=0.09$ for *NR5A2*, and $P_{\text{Fisher}}=0.06$ for *NR5A2*). However, because the genes are associated with at least two individual diseases after Bonferroni correction for the number of all genes and even after multiple-gene conditioned fine-mapping analysis (in the case of *NR5A2*, the signal is transcriptome-wide significant even for CD, UC and SCZ), and, moreover, the GWAS and TWAS correlation results clearly indicate a common GWAS/TWAS component, these genes in our view, represent excellent candidates of shared disease-genes associations. Nevertheless, to point out this limitation, we have added the following sentence to the new limitations section: "One limitation is that we could only use candidate genes from six TWAS loci for pathway analysis, so future, more powerful TWAS are expected to provide a more detailed picture here."

The number of comparisons is large. In fact, the total number of genes (3) that show significant association across traits and across brain and intestinal tissues seems low to me.

We admit that these three genes underwent very stringent filtering with (i) the Bonferroni correction for all protein-coding genes, (ii) the multiple-gene conditioned finemapping analysis, (iii) the GWAS-TWAS conditional analysis and (iv) the final condition for an association in at least one brain and non-brain tissue. Otherwise, less stringent filtering results in additional loci (Supplementary Table 13) for which the TWAS association signals could not be unambiguously assigned to a single gene at the respective locus; not even after multiple-gene conditioned fine-mapping (see conditional P-values for the loci/genes *SF3B1*, *INO80E*, *SGSM3* and *ZC3H7B* in Suppl Figures 8-11 and, e.g. for Crohn's disease, pages 6 (*SF3B1*), 8 (*SGSM3*), 9 (*ZC3H7B*) in file Supplementary_File_1.pdf), where more than one gene at a locus is still transcriptome-wide significant even after multiple-gene conditioning.

Since our goal was to uniquely identify a single gene per TWAS locus for each disease alone first, it is not unlikely that the overlap between diseases in the number of genes ($n=3$) for these unique genes is small and the gene enrichment analysis results no longer significant.

To what extent does this support the hypothesis of genetic overlap? (at a TWAS level).

In TWAS, similar to GWAS, it is generally difficult to describe the usually high genome-wide heritability estimate (e.g. by LDSC) by single individual genome-wide significant SNP marker signals alone. Current estimates suggest that common variants on an entire GWAS array may describe approximately 26% of the genetic variance in Crohn's disease, whereas the >230 established GWAS susceptibility loci for Crohn's disease describe about 13.1% of disease variance (PubmedID 26192919 and 28404137). Therefore, in our view, with these 3 overlap genes, we can find only the tip of the expected "polygenic" (shared) TWAS iceberg, but these are then very likely relevant (shared) candidate genes. We added the following sentence to the limitations section: "Another limiting factor of our study is that the heritability of our three (conservatively) selected gene-disease associations accounts for only a small fraction of the estimated h_{med}^2 and may present only the tip of the iceberg."

The authors refer to the gene expression heritability estimates, but although these are relevant, they form only a basis for significant TWAS association. These numbers are again reported in discussion, but do not provide insight into how much of the phenotypic/genetic variation in the traits is explained by these genes.

We agree with the reviewer that the ($h_{med-min}^2$, $h_{med-max}^2$) values of expression of individual genes are not informative of overall heritability. We have provided these values to illustrate the extent to which gene expression is genetically regulated and that TWAS are therefore a useful method for examining associations with these genes. At the single trait level, we further refer to the heritability mediated by gene expression as calculated by Yao et al. (PubmedID 32424349; MESC method) across GTEx tissues.

To better explain this to the reader we have included the following sentences in the new limitations section in the Discussion: "Since h_{med}^2/h_g^2 can vary considerably for different traits or diseases, we recommend that future TWAS studies additionally estimate h_{med}^2/h_g^2 (e.g. with the MESC software) to assess whether a TWAS can be expected to have a low or rather high number of gene-disease associations for the disease under study ... In addition, the proportion of shared h_{med}^2 between diseases is difficult to quantify and requires development of bivariate (for pairs of diseases) methods to estimate the shared h_{med}^2 ."

However, it was not possible for us to reliably calculate the mediated expression of a gene set of only 3 genes with the MESC program, so unfortunately we had to be satisfied with the expression heritability estimates of the gene candidates. To successfully calculate partitioned eQTL heritability with MESC, a certain number of loci or genes is required. This set of genes must be on the order of ~200 genes per gene set. In their Supplementary Note, Yao et al. (PubmedID 32424349) wrote in this regard, "Moreover, a key drawback of MESC is that it produces large standard errors for small gene sets and thus can only be applied to large gene sets with more than 200 genes, whereas other methods can analyze gene sets of any size. Thus, we propose MESC as a complementary approach rather than replacement for other pathway enrichment methods."

5) Throughout the paper, there are sentences where the information is so dense that they are difficult to read. I will give one example, but the entire manuscript should be checked. lines 229-231. I think that the authors imply that NR5A2 and SATB2 are associated with SCZ, CD and/or UC are in the same direction, while PPP3CA is associated in opposite directions. If this is true, what are the implications? (Tables and Figures actually suggest my interpretation was incorrect, still showing that the original sentence was difficult to read).

We apologize for the sometimes too short wording. We have deleted the sentence in lines 229-231 that was difficult to understand and further elaborated the content in the description of Table 2: "Increased predicted expression (indicated by a positive Z score) of *NR5A2* and *SATB2* is associated with increased risk SCZ and CD as well as SCZ and UC. Decreased predicted expression (indicated by a negative Z score) of *PPP3CA* is associated with increased risk for SCZ and CD." This means that the direction of effect of the association is the same for all diseases, respectively.

6) Lines 303-306: A gene can only have a significant TWAS association if there is expression (and variation in expression). Therefore, I am not sure this info is very relevant. It is using the same data as the TWAS, so findings are not surprising at all.

We agree that the sentence in lines 303 – 306, as well as the old subfigure Figure 4a (we now removed it), merely reflect (despite being a good quality control) what we already found in our TWAS reference data, and we have therefore deleted these sentences and the old subfigure Figure 4a to make room for the new paragraph in the Discussion regarding the limitations of our study. All information shown in Figure 4a is now contained in Supplementary Figure 12.

7) Lines 411-414: I am not very convinced or excited by this entire paragraph, as the authors overstate findings by ignoring the polygenicity of the traits. These sentences are especially problematic as the clinical implications described here are not supported sufficiently by their results. How much of the genetic/phenotypic variation in the traits is explained by the highlighted genes?

We agree that the results of clinical trials, mouse studies, and case reports of relapse of schizophrenia after tacrolimus use and the results of our studies do not suggest causality, although our results at least support the clinical and functional observations. For this reason, we have deleted lines 411-414, which refer to possible clinical implications of our results.

As described above under comment 4), the three genes describe only a fraction of the estimated polygenicity, similar to GWAS Risk SNPs. At the genetics level, as we now measured by a partitioned LDSC h^2 approach, our three candidate gene loci alone describe CD: 0.0015 (se=0.0003), UC: 0.0032 (0.0018) and SCZ: 0.0014 (0.0009) of heritability on the liability scale, with the genome-wide values being 0.24, 0.14, 0.22, respectively. However, similar to the GWAS association of genetic variants of a locus describing only a fraction of GWAS heritability, the locus can be indeed of great importance, e.g. the GWAS association of *IL23R* in Crohn's disease was one of the most important results of previous GWAS studies and another genetic support for further drug development (monoclonal antibody therapy) in Crohn's disease. From this point of view, we believe that the identification of novel candidate genes by means of TWAS is more important than the description of full heritability at the TWAS level.

8) The authors seem to have used outdated SCZ Summary stats although they do cite the latest one (Pardinas). The reason for this is unclear.

We apologize for the imprecise description. The GWAS summary statistics (35,476 cases, 46,839 controls) for SCZ are from the Pardinas 2014 study (PubmedID 25056061; as cited in Supplementary Table 1), which was the most recent and freely available GWAS dataset for SCZ at the time we started our study. In order to match the boundaries of the TWAS loci for our 3 genes in Table 2 with the boundaries of all known SCZ loci (see also Table 2), we additionally referred to the most recent list of 145 GWAS loci from Pardinas 2018 (PubmedID 31160808), which were later fine-mapped in this study there but for which no GWAS summary statistics were publicly available at that time we started our study.

9) The analyses for ADHD, MDD, and BD seem to be relatively underpowered. This should

be noted as a limitation. The authors could consider to use Howard et al. for the MDD GWAS (although these are not all clinical cases).

In our view and from the point of view of the number of GWAS samples used for TWAS analysis, our analysis of the GWAS datasets for ADHD (20,183 cases), MDD (59,851 cases), and BD (20,352 cases) had a high statistical power, for which reason these GWAS data have also already recently been investigated, for example, in Gamazon et al. Nat Genet 2019 (PubmedID 31086352) using the (single-tissue) TWAS method, whereas compared with the Gamazon et al. study (11,974 BD cases), we investigated a GWAS dataset for BD that was almost twice as large in our study.

We agree that in light of the new study by Yao et al Nat Genet 2021 (PubmedID 32424349, Supplementary Figure 20), in which Yao et al estimated the proportion of expression mediated heritability for 42 traits (including SCZ, UC, CD and "depressive symptoms") and 48 tissues from GTEx, that at least for MDD (ADHD and BD were not examined in this study) the statistical power in a TWAS is likely to be very low (estimated proportion of expression mediated heritability $h_{med}^2/h_g^2 = -0.04$, $sd = 0.069$, $P = 0.26$ for "Depressive symptoms", PubmedID 32424349, their Supplementary Table 3) that a large number of genes will appear as gene-disease associations in a TWAS for MDD (see also Figure 2a in our manuscript), in contrast to SCZ, UC and CD with a significant estimated proportion of expression mediated heritability (h_{med}^2/h_g^2) of about 11.5%, 38.2% and 26.7%, respectively (PubmedID 32424349, Supplementary Table 3). In order to make this limitation clear to the reader, we have included the following sentence in the limitations paragraph in the manuscript: "Since the proportion of expression-mediated heritability (h_{med}^2/h_g^2) can vary considerably for different traits or diseases, we recommend that future TWAS studies additionally estimate h_{med}^2/h_g^2 (e.g. with the MESOC software PubmedID 32424349) to assess whether a TWAS can be expected to have a low or rather high number of gene-disease associations for the disease under study."

As noted in comment 8) above for SCZ, at the time we began our study, the GWAS dataset of 59,851 MDD cases from the Wray et al 2018 (PubmedID 29700475) was the largest freely available GWAS dataset of physician-diagnosed MDD cases. As correctly anticipated, we intended to limit our TWAS study to GWAS data from physician-diagnosed cases only, as with the other six diseases of our study.

10) From the main text, it seemed that the 3 key genes were associated across brain and intestinal tissues. From Table 2 it becomes clear that the associations for SCZ only exist in brain. This seems to contradict the main conclusion of the authors that these pleiotropic genes influence immune and psychiatric traits through their role in brain and intestinal tissues?

The three genes in Table 2 are transcriptome-wide significant ($P_{conditional} < 3.20 \times 10^{-6}$) in the same brain or non-brain tissue for at least one immune disease and one psychiatric disease, where at least one of the diseases must also show another transcriptome-wide significant signal for another brain or non-brain tissue (i.e. brain tissue if first signal occurred in a non-brain tissue or *vice versa*). This ensures that it is a pleiotropic signal for brain and non-brain tissue. Moreover, this can be used to express that, for example, a particular gene in CD exerts its pathological function mainly in the intestine and for SCZ mainly in the brain.

From a statistical point of view, we think it is too stringent to require that each of the 3 genes has to be transcriptome-wide significant for at least one immune disease and one psychiatric disease in the same brain and in the same non-brain tissue. For example, the *PPP3CA* gene is transcriptome-wide significant for CD in "Colon transversum" and "Putamen basal ganglia", is transcriptome-wide significant for SCZ in "Putamen basal ganglia" and has a P value of 0.007 for SCZ in "Colon transversum" after multiple-gene conditioned analysis (see Supplementary_File_1.pdf, page 16 and Supplementary Table 3). This weaker association

signal for SCZ in "Colon transversum" could also be explained by the fact that we have a lower statistical TWAS power to detect it because pleiotropic effects, on average across all tissues, are likely to be weaker for SCZ (h_{med}^2/h_g^2 of about 11.5% for SCZ across 48 GTEx tissues, see answer 9) above) as compared to CD (h_{med}^2/h_g^2 of about 26.7% for CD across 48 GTEx tissues, see answer 9) above).

To make this clear to the reader, we have rephrased the sentence from lines 221-226 as follows: " In summary, the three genes met the transcriptome-wide significance threshold of 3.20×10^{-6} in the same brain or non-brain tissue for at least one immune disease and one psychiatric disease, where at least one of the diseases must also show another transcriptome-wide significant signal for another brain or non-brain tissue (brain tissue if first signal occurred in a non-brain tissue or *vice versa*)."

11) Overall, the quality of figures and tables is good. However, Figure 3 is difficult to understand. Even after reading the legend several times, I still do not shown what information is shown below the manhattan plots (green circles and green bars). It should be explained in the figure itself. The other information was clear in the figure labels, but not so much in the legend. Also, in figure 3, the * (TWAS gene eQTLs are difficult/impossible to detect.

We are very grateful for the suggestions for improving the readability of Figure 3. In the new version of Figure 3 the legend now also describes the green circles and bars which were previously described in the text of the figure. The stars in the GWAS plot correspond to the green circles (TWAS gene eQTLs) and lie directly vertical above the green circles, but, due to the abundance of GWAS SNPs, are only well visible if they also have a small GWAS P-value and are therefore only of interest in this case at the GWAS level.

12) Figure 5: how much of the genetic/phenotypic variance is explained by this pathway? (or the genes in this pathway). Could the authors conduct a more formal enrichment analysis?

The genetic heritability on the liability scale of the genes in this pathway was 0.0145 (s.e. 0.0037), 0.0114 (s.e. 0.0038), 0.0036 (s.e. 0.0022) for CD, UC and SCZ, respectively. The genome-wide values were: 0.24, 0.14 and 0.22 as already mentioned in comment 7). This was calculated by a partitioned LD-score using LDSC with the 55 genes of the pathway (Figure 5) and a 1 cM frame. The purpose of the enrichment analysis in paragraph "Calcineurin-dependent NFAT and Wnt signaling as shared signaling pathways for IBD and SCZ" was to prioritize candidate genes in particular in the additional GBJ-significant loci (*INO80E*, *SGSM3* and *ZC3H7B* from Supplementary Table 13). Calcineurin signaling, possibly via Wnt and via NFAT, is thus a common theme for our TWAS candidate loci. We do not believe that a more formal enrichment analysis based on ontologies of all TWAS hits for the respective features would be informative for finding shared ontology terms, as they would be listed behind the major disease-relevant terms. As the reviewer correctly noted above, both IBD and SCZ are polygenic and therefore multiple pathways and ontologies are involved. To make the purpose of the enrichment analysis clearer to the reader, we have added the following sentence (in context of the additional GBJ-significant loci (*INO80E*, *SGSM3* and *ZC3H7B*)) in the Results section: "We assumed that these genes, which according to previous findings have nothing to do with IBD or SCZ (Supplementary Box 2), may be masking the causative gene of the locus. To test whether any of the secondary gene candidates could be linked to *NR5A2*, *SATB2* and *PPP3CA*, we performed gene set enrichment analysis as implemented in EnrichR using the NCATS BioPlanet pathway resource ..."

To show that the enrichment analysis is based on only a few loci and genes, we have added to the limitations section: "Together with the results of our gene set enrichment pathway analysis, we showed it is likely that genetic variation mediated by gene expression in the Wnt signaling pathway and NFAT activation by calcineurin is associated with both SCZ and IBD.

One limitation is that we could only use candidate genes from six TWAS loci for pathway analysis, so future, more powerful TWAS are expected to provide a more detailed picture here.”

Minor comments:

- Line 111: “Gene expression imputation was conducted using GWAS summary statistics”. I’m not sure if I would refer to this as imputation as no raw genotype data (individual-level) were available

We changed the wording from “gene expression imputation” to “using imputation models” throughout the text.

- Line 114: combining N controls across traits is not very informative.

We removed the total number of controls from Title, Abstract and Results section.

- Line 182: reference to CD4+ T-cells is confusing, was this not a genome-wide analysis?

We performed pair-wise trait correlation analyses on the single tissue level (Single-tissue_{marginal}) and also the cross-tissue level (GBJ_{marginal}). The strongest positive correlation across psychiatric and immune phenotypes on the single tissue level was observed for tissue “CD4+ T-cells”.

- Line 187-190: Can the authors statistically validate this statement?

While the correlation values between disease pairs at the TWAS level are on average about half as large as the correlation values at the GWAS level, we admit that we cannot, however, provide an accurate estimate of the extent to which the common genetic variants for particular combinations of disease pairs contribute to the common correlation at the TWAS level. For this purpose, a specific bivariate version of the current MESC software (PubmedID 32424349) would be required, which is currently not available. As noted in comment 4), we have discussed this in the new limitations section of the Discussion. In addition, we have now rephrased this sentence as follows: “... suggesting that part of the effects of genetic variants shared between psychiatric and immune phenotypes are likely to be mediated by gene expression”.

- Line 212: Why was the number of genes 171 for the MR analysis and 288 for this analysis?

The number of valid pairwise tests in the MR analysis was only 171 (i.e. 552 (number of all possible pairs; the order of pairs is important) minus 381 (number of non-valid analyses, where the number of instruments was lower than 10)) since at least 10 significant, independent association signals were necessary as instruments to obtain reliable regression results in MR analysis, see our Methods section and Zhu et al 2019 (PubmedID 29335400). On the other hand, in the gene overlap analysis we have tested 288 possible pair-wise tissue-specific combinations of psychiatric and immune phenotypes where the order of the pairs was not important.

- Line 247: typo, GTCA should read GCTA

We have corrected the typo.

- Line 277: How can a gene be implicated by a non-genetic study?

We have corrected the wording, because "non-genetic" studies should mean functional and expression studies.

- Line 572-573: I think this would also need to explained in the GJB test section

We now deleted this sentence because this had no influence on the Spearman correlation analysis.

- The font size and quality of Figure 2B should be improved

This seems to be only a problem with the online submission conversion to PDF. In our Word manuscript file this image has a very high resolution.

Reviewer #1 (Remarks to the Author):

The authors have successfully addressed my comments.

Reviewer #2 (Remarks to the Author):

I thank the author for their extensive responses to my earlier comments and I am satisfied with their answers.

As a minor note, the 2014 Schizophrenia study is from Ripke et al. not Pardini et al. so please check whether it's correctly cited in the manuscript.

From the rebuttal letter

"We apologize for the imprecise description. The GWAS summary statistics (35,476 cases, 46,839 controls) for SCZ are from the Pardini 2014 study"